# Benchmarking Progress to Infant-Level Physical Reasoning in AI

Luca Weihs,[1]   Amanda Rose Yuile,[2,∗]  Renée Baillargeon,[2]
Cynthia L Fisher,[2]   Gary Marcus,[3]   Roozbeh Mottaghi,[1]   Aniruddha Kembhavi[1]

[1] *Allen Institute for AI,* [2] *University of Illinois at Urbana-Champaign,* [3] *New York University*

*{lucaw, roozbehm, anik}@allenai.org*
*{amandah3, rbaillar, clfishe}@illinois.edu*
*{gfmarcus}@gmail.com*

**Reviewed on OpenReview:** *https://openreview.net/forum?id=9NjqD9i48M*

**Data, Code, and Videos available at:** *https://allenai.org/project/inflevel*

## Abstract

To what extent do modern AI systems comprehend the physical world? We introduce the open-access Infant-Level Physical Reasoning Benchmark (INFLEVEL) to gain insight into this question. We evaluate ten neural-network architectures developed for video understanding on tasks designed to test these models' ability to reason about three essential physical principles which researchers have shown to guide human infants' physical understanding. We explore the sensitivity of each AI system to the continuity of objects as they travel through space and time, to the solidity of objects, and to gravity. We find strikingly consistent results across 60 experiments with multiple systems, training regimes, and evaluation metrics: current popular visual-understanding systems are at or near chance on all three principles of physical reasoning. We close by suggesting some potential ways forward.

## 1   Introduction

As artificial intelligence increasingly begins to power automated vehicles and robots' actions, generalization to the infinite variety of real-world environments requires that these AI systems reason use a reliable understanding of the physical world. For instance, an automated vehicle that does not understand that permanence is a general attribute of all objects may not react appropriately when a novel object becomes occluded on a roadway. While there is an active community of researchers pursuing this direction (Lake et al., 2017), most current AI work is focused on end-to-end learning with deep-learning architectures (Le-Cun et al., 2015). As they have grown in size and leveraged increasing amounts of data, they have shattered records across numerous computer-vision benchmarks, including image classification  (Zhai et al., 2021),[1] object detection (Xu et al., 2021),[2] action recognition (Ryoo et al., 2021),[3] visual question answering (Wang et al., 2021),[4] and complex reasoning tasks in synthetic environments (Yi et al., 2018).[5] And yet, even these remarkable results give researchers little confidence that today's best models have anything approaching human-level world understanding (Serre, 2019; Yuille & Liu, 2021).

Developmental psychologists have shown that by 4.5 months of age, infants correctly reason about objects' displacements and interactions in many different physical events. The presence of this ability at such a young

---

∗Work completed, in part, while an intern at the Allen Institute for AI.

[1]>90% Top-1 accuracy at ImageNet classification
[2]60 box AP at COCO object detection
[3]85% Top-1 accuracy at Kinetics-400 action recognition
[4]80% Top-1 accuracy at visual question answering
[5]>99% accuracy and >90% Ave-per-q at the diagnostic CLEVR and CLEVRER datasets

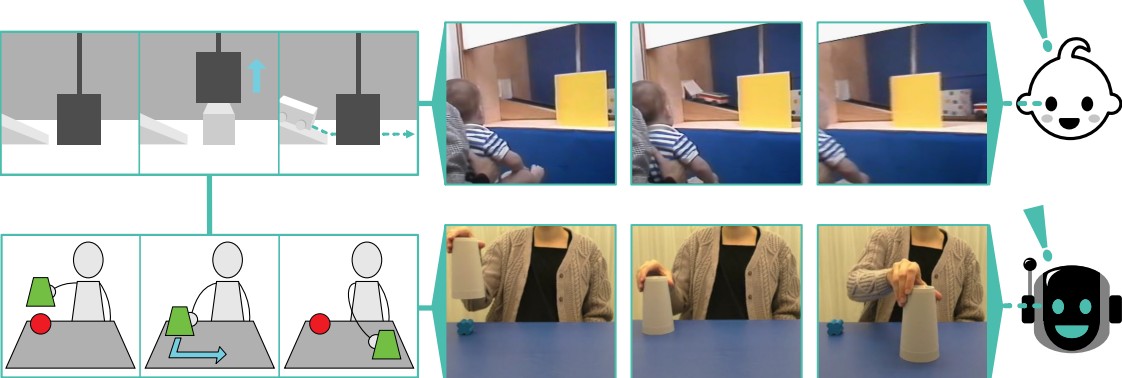

Figure 1: **Evaluating physical-reasoning abilities of AI models.** Inspired by studies performed with infants, INFLEVEL asks AI models if they are surprised by videos of physically plausible and implausible events. **Top**: infants are surprised by a Solidity violation where a toy car rolls through a box (Baillargeon, 1986). **Bottom**: schematic and example from INFLEVEL; models that understand Solidity should be surprised that the object does not move with the cover, as though it somehow passes through the back of the cover.

age has led to the formulation of the *core-knowledge hypothesis*, which states that infants are born with an innate understanding of core physical-reasoning principles (Spelke et al., 1992). These principles are now thought to include Persistence (all other things being equal, objects persist as they are in time and space), Inertia (absent an external force, moving objects follow straight trajectories and stationary objects remain stationary), and Gravity (objects fall unless supported). Persistence is composed of five subcategories: Continuity, Solidity, Cohesion, Boundedness, and Unchangeableness, for a total of seven principles (Lin et al., 2022). We capitalize the names of these principles to emphasize that these terms have meanings within developmental psychology that may differ slightly from everyday use. While infants cannot identify 1000 object categories or answer complex questions about images, their innate understanding of these core principles is the foundation upon which remarkable capacity for physical understanding is built. We believe that assessing a video-understanding model's knowledge must include a direct evaluation of its understanding of these core principles. If a model's representation of the world failed to embed the understanding that objects should fall when unsupported, then we would have serious concerns that this model was performing any meaningful physical reasoning, regardless of how strongly it performed on a computer vision benchmark.

To assess contemporary AI systems' ability to reason about some of these essential aspects of the physical world, we present the Infant-Level Physical Reasoning Benchmark (INFLEVEL). We focus on three principles that have been extensively studied, Continuity, Solidity, and Gravity, modeled on past experimental work in developmental psychology (Lin et al., 2022; Baillargeon, 2008; Baillargeon et al., 1985; Luo et al., 2009; Needham & Baillargeon, 1993). These principles enable infants to correctly interpret a myriad of simple object interactions and thereby they provide a good objective for study. These principles can also all be studied with a consistent experimental setup (a single human manipulating two objects on a table). In this prior work, infants' understanding is typically assessed using the violation-of-expectation (VoE) paradigm (Lin et al., 2022) in which infants are presented with a physically implausible event, which violates the principle under investigation, and a physically plausible event, which conforms to this principle. With appropriate controls, greater surprise at the implausible event, with looking time as the canonical surrogate measure quantifying surprise, is taken to indicate that the child brought to bear the relevant principle to form expectations about how the events would unfold and subsequently detected the violation in the physically implausible event. In physical-reasoning experiments, infants tend to look longer at unexpected outcomes, somewhat analogous to an adult having a double-take upon seeing a magic trick. In essence, the premise is that any agent that can understand the world should be able to discriminate between ordinary events (*e.g.*, inert objects falling towards the earth) and unusual events (*e.g.* inert objects floating in midair, unsupported). With this VoE paradigm in mind, INFLEVEL contains two video datasets, INFLEVEL-LAB and INFLEVEL-SIM, with videos involving physically plausible and implausible object interactions (see Fig. 1).

INFLEVEL-LAB contains ~5,700 videos filmed in an infant-cognition lab, divided into 3 subsets focusing on Continuity, Solidity, and Gravity. These principles have the advantage that they do not require elaborate experimental set-ups, even when creating physically implausible videos (see Sec. 3). We propose quantifying a model's surprise using metrics designed for out-of-domain (OOD) detection. This novel perspective on quantifying surprise in artificial models allows us to evaluate any visual model that produces a vector representation of an input video. We pair our natural video benchmark INFLEVEL-LAB with INFLEVEL-SIM: ~75,000 videos collected using the robotic-arm enabled agent within the AI2-THOR environment (Kolve et al., 2019; Ehsani et al., 2021). Mirroring its natural video counterpart, INFLEVEL-SIM includes videos of plausible and implausible interactions between objects but contains an order of magnitude more videos due to the inexpensive nature of creating videos within a simulated environment. Unlike related benchmarks (see Sec. 2), our inclusion of a real-video dataset enables us to test models trained on natural-video datasets with minimal domain gap. Likewise, as we also wish to evaluate embodied agents (generally trained in synthetic environments like AI2-THOR), our inclusion of INFLEVEL-SIM reduces the domain gap in evaluating such agents.

INFLEVEL has several unique design considerations that break convention with usual benchmarks in computer-vision: **(1) Evaluation only**: INFLEVEL is designed to be *used only for evaluation*. As discussed in Sec. 3, allowing training (even of a linear probe) may result in evaluation metrics confounding physical understanding with feature extraction and matching. This result underlines the subtlety of evaluating the expectations embedded in model representations. We stress that INFLEVEL is designed to assess the physical reasoning abilities of already trained models; we do not allow for any training or fine-tuning on our data as this would lead us to evaluating "what a model can learn" and not "what a model currently understands." Disallowing training makes INFLEVEL challenging; this is by design and is how infants are evaluated. **(2) Easy for humans**: As INFLEVEL builds on infant experiments, adults evaluated on it obtain near 100% accuracy in our exploratory experiments. Given this "triviality," a model's failure emphasizes the substantial gap between the physical-reasoning capabilities of infants and artificial agents. **(3) Controlled**: The events depicted in INFLEVEL videos are highly controlled and mirror events shown to infants. This tight control opens the door for designing heuristics that can obtain high performance. We emphasize that INFLEVEL is intended to be a diagnostic methodology for studying the physical knowledge embedded in learned representations: a poor performance suggests a fundamental failure of the representation to embed physical principles; high performance, on the other hand, must be understood in the context of how the model was trained and the inductive biases it employs.

We evaluate a range of state-of-the-art (SoTA) video models trained on popular large-scale datasets (*e.g.*, HowTo100M, Kinetics400, SSv2, etc. (Sigurdsson et al., 2016; Miech et al., 2019; Kay et al., 2017; Goyal et al., 2017; Damen et al., 2018). These models perform poorly (marginally above random chance) when presented with the Continuity violations; on Solidity and Gravity violations, they perform even worse. On INFLEVEL-LAB, all models attain near-random performance across all principles suggesting that learning from embodied interaction may require more than training agents for task completion.

In summary, our contributions include: (i) INFLEVEL, a benchmark containing thousands of natural and synthetic videos designed to evaluate AI systems' understanding of core physical principles, (ii) a proposal to use OOD detection as a flexible proxy for surprise in artificial agents that does not require retraining, and (iii) exhaustive experimentation demonstrating that SoTA video understanding and embodied models find INFLEVEL exceptionally challenging. Data and code will be made publicly available.

## 2 Related Work

**Physical Reasoning in Infants.** The study of physical reasoning in infants has a long history dating to the seminal works of Piaget (Piaget, 1952; 1954). Piaget's experiments led him to hypothesize that infants' physical reasoning abilities were relatively crude, even lacking object permanence until 8-9 months of age. In part due to the introduction of the *violation-of-expectation* (VoE) paradigm (Baillargeon et al., 1985), these hypotheses have been largely overturned. Infants have surprisingly sophisticated physical-reasoning capabilities; for instance: 2.5-month-olds expect objects to become visible when moving between occluders (Aguiar & Baillargeon, 1999), 4.5-month-olds expect inert objects to fall when unsupported (Needham

& Baillargeon, 1993), and 3.5- to 4.5-month-olds expect objects to follow straight-line trajectories and to collide with hidden objects (Baillargeon, 1986; 1987). See the chapter of Lin et al. (2022) for an in-depth survey of these advances and for a discussion of the *core-knowledge hypothesis* which suggests that infants are imbued, at birth, with a "skeletal framework of core physical principles" which they leverage to understand the world. Inspired by this work, we design INFLEVEL to test artificial agents' understanding of a subset of these core physical principles.

**Physics Reasoning Benchmarks and Models.** There is a substantial history of work studying physical reasoning in artificial models (Lake et al., 2017). Here we will highlight the pieces most relevant to INFLEVEL. In INFLEVEL, agents are required to reason about objects that are hidden behind occluders, inside containers or tubes, or under covers. Relatedly, researchers have considered knowledge-base systems for reasoning about container manipulation (Davis et al., 2013), built generative models of containers' contents to guide active search (Wong et al., 2013), designed systems to reason about containment in videos (Liang et al., 2016) and to track occluded objects through long temporal sequences (Liang et al., 2018), and considered parse graphs as a means by which to track object states from multiple views with potential occlusion (Yuan et al., 2020). Other works have studied building models of intuitive physics by using inverse graphics to approximate an observed scene inside of a game engine (Wu et al., 2017), Xu et al. (2019) introduce DensePhysNet uses input depth maps to predict object state changes after actions in a self-supervised manner, and the Dynamic Concept Learner (DCL) uses object-centric inductive biases, a learned dynamics model, and graph NNs to obtain SoTA performance on the CLEVRER benchmark (Chen et al., 2021). These works present new methodology and models while our focus is on designing a benchmark to evaluate existing models, potentially trained for unrelated end tasks.

Of recently proposed physical reasoning benchmarks, most similar to INFLEVEL are IntPhys (Riochet et al., 2018) and two unnamed datasets from Piloto et al. (2018) and Dasgupta et al. (2021), which we call PilotoVoE and DasguptaVoE respectively. These benchmarks explicitly reference the VoE paradigm and evaluate models by testing their ability to discriminate between physically plausible and implausible events. Our work differs from theirs along two critical dimensions. First, these other benchmarks are set within stylized simulations, with simple geometric objects that move without any apparent external force acting upon them. In contrast, we include natural videos in INFLEVEL-LAB, use diverse household objects, and always include an agent who explicitly manipulates objects. Second, these benchmarks are designed with a different use-case in mind: they all include large datasets of plausible videos meant to be used to train models to physically reason explicitly. INFLEVEL is evaluation-only: we have no training dataset by design. This is because we are interested in testing the capabilities of models *that have already been trained*, to understand whether these models have learned to reason about core principles through their normal course of training. We also note that we focus on very simple physical violations; by contrast, violations in IntPhys are sufficiently complex that even humans obtain high error rates (up to an 47% error rate in the most challenging setting!). In recent concurrent work, Piloto et al. (2022) have released the Physical Concepts dataset, a dataset similar to PilotoVoE but with richer coverage of physical concepts; they also introduce a "PLATO" model which, given preprocessed information in the form of groundtruth semantic masks and object indices, attains impressive accuracy on their dataset. Our work differs from this work just as it does for the original PilotoVoE benchmark as described above. Additionally, our benchmark could not be directly tested on their model as is, since their system, unlike the systems that we did examine, does not work purely from pixel-based input.

Several other benchmarks have recently been proposed. Physion (Riochet et al., 2018) is set in the simulated ThreeDWorld environment (Gan et al., 2020) and studies physical understanding by requiring models to predict whether two objects will come into contact in the future. PHASE (Netanyahu et al., 2021) presents 2D animations of abstract social interactions with associated social recognition and prediction tasks. AGENT (Shu et al., 2021) generates simulated 3D videos to probe understanding of how agents plan and execute efficient, goal-directed actions. BIB (Gandhi et al., 2021) tests agents' capacity for predicting other agents' behavior using simulated 2D/3D videos. CLEVRER (Yi et al., 2020), built upon CLEVR (Johnson et al., 2017), tests models' ability to answer questions about potential collision events. Physics 101 (Wu et al., 2016) presents natural videos to study how models may learn physical attributes from unlabelled videos. We also highlight that recent years have seen the proliferation of high-fidelity, physics-enabled, sim-

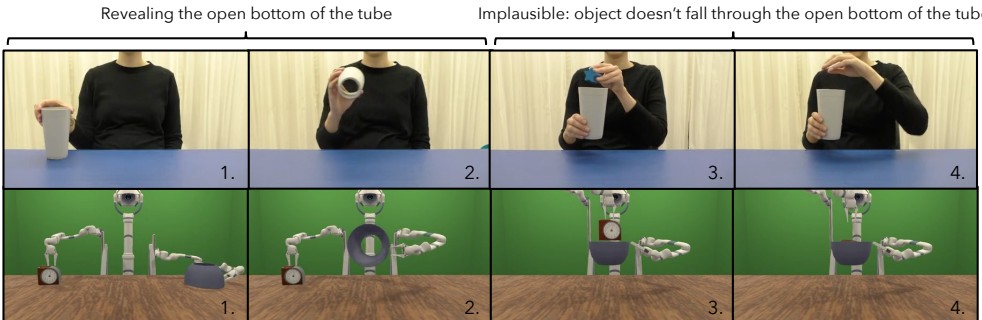

Figure 2: **Implausible Gravity examples**. Four frames taken from Gravity trials in InfLevel-Lab and InfLevel-Sim. The operator shows that the primary object is a tube and then drops an object into it. The object fails to fall through the open bottom of the tube, creating an implausible outcome.

ulated worlds (Kolve et al., 2019; Xiang et al., 2020; Szot et al., 2021; Li et al., 2021) which allow explicit embodiment and learning-from-interaction.

## 3 InfLevel: The Infant-Level Physical Reasoning Benchmark

InfLevel is designed to evaluate the core physical-reasoning abilities of existing video-understanding models. To accomplish this goal, InfLevel must balance two competing goals:

**1. Simplicity.** To ensure that we are testing models' core physical understanding, the videos in InfLevel should be simple, closely replicating the lab settings used in infant studies. While ideal models would be both robust to distraction and able to reason precisely in complex multi-object scenes, such complexity would only add noise to our desired signal: it would be challenging to know if a model's failure was due to a lack of physical understanding or, instead, to confusion in the presence of distractors.

To achieve simplicity, we make several design decisions. All events depict interactions between precisely two objects, a primary and a secondary object. The primary object is a container, a cover (an upside-down container), or a tube (a bottomless container); the secondary object is placed into the container, under the cover, or into the tube. All objects are simple, household objects (e.g., cups, bowls, blocks, or crayons) that infants may encounter in everyday life (see Fig. 5). The primary and secondary objects are manipulated on a table by an *operator* (a human in InfLevel-Lab and a robotic agent in InfLevel-Sim). The interaction between the two objects is either physically plausible or violates a core physical principle (see Fig. 2).

**2. Diversity.** We must have enough visual diversity in our videos so that high performance is not simply a product of a model accidentally latching onto some statistical correlation present across videos. To diversify our videos, we use a large collection of objects, provide three camera angles, and vary the direction in which trials are performed. See App. Fig. A.1 for an example of such diversity. In InfLevel-Sim, we also change the texture and color of the table and background programmatically; in InfLevel-Lab, the human operator also varies their clothing. Importantly, our strategies for increasing diversity are cosmetic and do not require agents to reason about additional objects or new physical concepts.

### 3.1 Experimental Designs

We now discuss the physically plausible and implausible event videos used to test models' understanding of Continuity, Solidity, and Gravity. In each case, we include two plausible-implausible pairs of trials.

**Continuity.** The principle of Continuity, a corollary of Persistence, states that an object should not spontaneously appear or disappear (Lin et al., 2022). Infants as young as 2.5 months of age are able to reason about Continuity violations. For instance, they detect a violation when an object that is passing behind two occluders positioned a short distance apart fails to become visible between them (Aguiar & Baillargeon, 1999). Similarly, they detect a violation when a cover is lowered over an object, slid to the side, and then

lifted to reveal no object (Wang et al., 2005). Our Continuity trials involve covering events, and each trial has three parts (separated by short, 0.5s, cuts):

1. *Secondary Object Familiarization* - The operator picks up the secondary object, moves it to the frame's center, and then places it back onto the table.

2. *Primary Object Familiarization* - The operator picks up the primary object (always a cover in these trials), rotates its opening toward the camera to show that it is empty, and then returns it to the table.

3. *Continuity Test* - The operator picks up the cover, lowers it over the secondary object, slides it across the table, and finally lifts it into the air. In a physically plausible event, the secondary object is revealed when the cover is lifted. We label this VV (the secondary object is visible at the start and end). In a physically implausible event, the secondary object is absent when the cover is lifted, labeled VI (visible-to-invisible). See Fig. 3 for schematics of VV and VI.

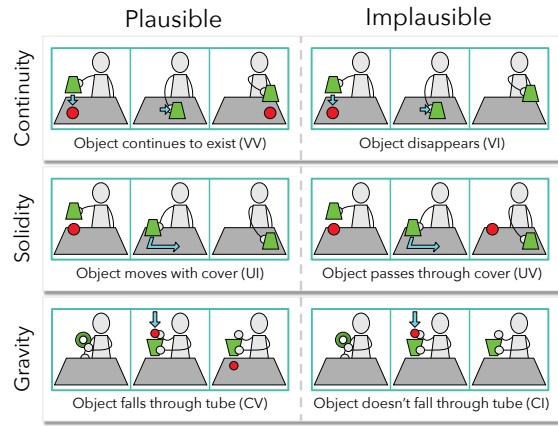

Figure 3: **Illustrations of (VV,VI), (UI,UV), & (CV,CI) events.** In Continuity, Solidity, and Gravity trials respectively.

Our Continuity trials also include II and IV, where no secondary object is present at the beginning. Evaluations now reverse, with IV being physically implausible (this is also a Continuity violation), and II being physically plausible. These additional trials ensure that the high-level statistics of our trials are well-balanced, *e.g.* there is not a clear bias where all plausible videos include two objects in the final frame.

**Solidity.** Another corollary of Persistence is Solidity, which states that two solid objects should not be capable of passing through one another (Lin et al., 2022). By 3.5 to 4.5 months of age, infants are surprised when a screen rotates through the space occupied by an object in its path (Baillargeon, 1987), when a toy car rolls through the space occupied by an object (Baillargeon & DeVos, 1991), or when an object is lowered into a container that is then moved forward and to the side to reveal the object standing in its initial position (Hespos & Baillargeon, 2001). Our Solidity trials' primary object is a cover (U); additional trials involve a clipped cover (C), which has been modified so that it is open at the back and functions as an occluder, see Fig. 4. Each Solidity trial has two parts (separated by short cuts):

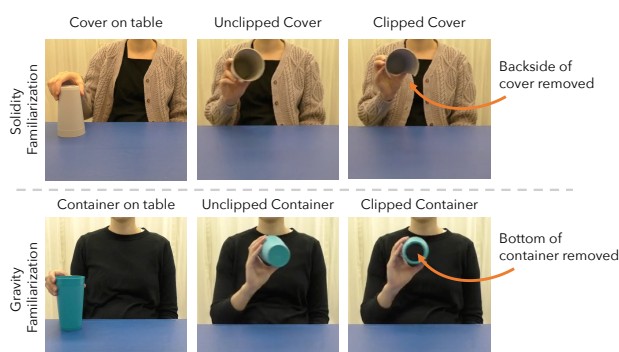

Figure 4: **Clipped and unclipped objects**. In the Solidity and Gravity trials, primary objects can be either clipped or unclipped. In the Solidity trials (top), covers have no back; in the Gravity trials (bottom), clipped containers have no bottoms.

1. *Primary Object Familiarization* - The U cover initially rests on the table. The operator lifts it, rotates its opening forward to show that it is empty, and then returns it to its original position.

2. *Solidity Test* - The U cover and secondary object stand next to each other on one side of the table. The operator lifts the cover, lowers it over the object, slides the cover forward, and finally slides it to the other side of the table. In the physically implausible event, the object is visible in its original position, as though it did not move along with the cover (UV); in the physically plausible event, the object is not visible (UI), as though it did move along with the cover. See Fig. 3 for schematic examples of the UI and UV cases.

Additional trials use the clipped cover (C); the missing back of the cover is revealed in the familiarization event when the cover is rotated forward. Plausibility now reverses, with invisible-at-the-end (CI) now being physically implausible (this is an inertia violation, as the clipped cover cannot exert a force on the object), and visible-at-the-end (CV) being physically plausible. These additional trials ensure that our Solidity

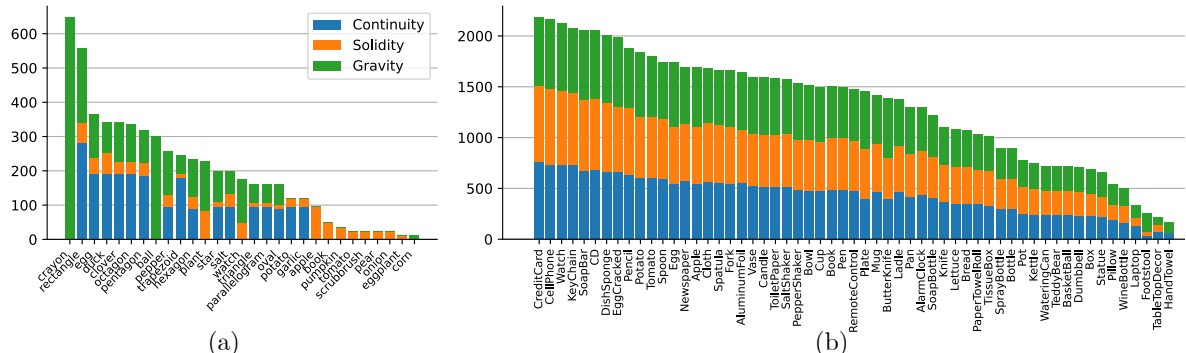

Figure 5: **Object type statistics.** Number of times various secondary object types appear in videos for (a) InfLevel-Lab and (b) InfLevel-Sim.

evaluation set is balanced: there are equal numbers of plausible/implausible videos where secondary objects are/aren't visible in the original positions.

**Gravity.** This principle states that unsupported objects should fall. By at least 4.5 months of age, infants are surprised when objects float unsupported (Needham & Baillargeon, 1993; Luo et al., 2009). Our trials involve a clipped container (C) that has been modified so that it is open at the bottom and functions as a tube; additional trials include a standard, unmodified container (U), see Fig. 4. These trials occur in two parts (separated by short cuts):

1. *Primary Object Familiarization* - The C container rests on the table. The operator lifts it, moves it to the center of the frame, rotates it to show that it is bottomless, and then returns it to its original position.

2. *Gravity Test* - To start, the C container and secondary object rest next to each other on one side of the table. The operator lifts C in one hand and moves it to the center of the frame (because C is upright, its open bottom cannot be seen). The operator then lifts the secondary object, lowers it into the opening of the container, and then lets go of the object. In the physically plausible event, the object falls through the open bottom of the container and onto the table (CV). In the physically implausible event, the object remains in the C container, even though there is no solid surface to block its fall and it is, therefore, unsupported (CI). See Fig. 3 for schematic examples of the CV and CI cases.

Additional trials use U, whose solid bottom can be seen in the familiarization event when the container is rotated forward. Events now reverse, with visible at the end (UV) being implausible (this is a Solidity violation, as the object appears to pass through the bottom of the container), and invisible at the end (UI) being plausible. As for Solidity, these additional trials help balance the evaluation set.

**Concept Overlap.** While the above trials would ideally evaluate each principle in isolation, it is impossible, even in infant studies, to completely disentangle all physical reasoning principles in a single event. Our design choice to use similar events with hidden objects across trials means that all of our violations involved Continuity to some extent: to detect them, an agent had to assume that the secondary object continued to exist after it became hidden under the cover (Continuity and Solidity violations) or inside the tube (Gravity violation). That issue aside, each violation focused primarily on one principle (*i.e.*, the hidden object appeared to wink out of existence, to pass through a solid surface, or to remain stable without support), making it possible to compare responses across principles.

### 3.2 InfLevel-Lab: Natural Videos

InfLevel-Lab consists of 5,772 videos filmed within an infant-cognition lab (2268 Continuity, 912 Solidity, and 2592 Gravity videos). We use 54 unique secondary object instances from 29 categories (see Fig. 5a) and 20 unique primary object instances from 3 categories (bowls, cups, and pots). Videos were filmed at 29.97 FPS in 1280×720 resolution. Continuity, Solidity, and Gravity trials take approximately 21, 16, and

14 seconds to complete respectively. Further details about this dataset, including a description of how we film events that appear physically implausible without visual special effects, can be found in Appendix A.

### 3.3 INFLEVEL-SIM: Synthetic Videos

While our primary interest lies in understanding the capabilities of models trained on real-world videos, researchers have argued that embodiment and interaction are critical for developing truly intelligent agents (Fridman & Malik, 2020) and presented evidence that developing general AI requires embodiment (Kremelberg, 2019). Also, recent work has shown some evidence that representations learned from gameplay in simulated environments have desirable properties (Weihs et al., 2021). As a large collection of the research in embodied AI is done in simulation (as real-world training is expensive and slow), we developed the INFLEVEL-SIM benchmark to evaluate embodied agents' physical-reasoning capabilities.

INFLEVEL-SIM consists of 75,336 videos collected in the simulated AI2-THOR environment (Kolve et al., 2019) using the robotic-arm-enabled ManipulaTHOR agent introduced in (Ehsani et al., 2021). We chose to use AI2-THOR as it, for our purposes, provided a useful feature set: physics-based arm interactions, high visual fidelity, material randomization, and a diverse set of object instances. Alternative potential embodied-AI simulators include iGibson 2.0 (Li et al., 2021) and Habitat 2.0 (Szot et al., 2021). To record our trials of interest, we had to extend the existing arm functionality to include new action types (*e.g.* arm rotation) and to add a second arm for use in our Gravity trials where two objects must be manipulated at once (see Fig. 2). These 75k videos are approximately evenly distributed across Continuity, Solidity, and Gravity trials (each having >24k videos); they involve 574 unique secondary objects from 57 categories (see App. Fig. 5b) and 167 unique primary objects from eight categories. Videos were recorded at the same FPS and resolution as in INFLEVEL-LAB and the Continuity, Solidity, and Gravity trials take approximately 30, 23, and 25 seconds to complete respectively. See App. B for more details.

### 3.4 Evaluation

We wish to evaluate models that have already been trained on external data sources (*e.g.* Kinetics400) without making strong assumptions on how they structure their output. To this end, we will assume merely that a given model $f$ can produce a vector representation of an input video $v$. A common strategy used to measure the quality of representations is via linear-probing: the model $f$ is frozen and a linear decoder is trained using the representation from $f$ as input to accomplish some auxiliary task. This approach works well in most settings as it captures a common use-case for representation learning methods: low-sample transfer learning. Unfortunately, as we discuss in detail in Appendix E.1, even training a simple linear decoder on INFLEVEL can result in conflating physical understanding with other related quantities.

For this reason, INFLEVEL is an evaluation-only benchmark: no training on INFLEVEL is allowed. This has an unfortunate implication we must overcome: models that wish to report scores on INFLEVEL must be able to provide a scalar "surprise" score for every input video. This requirement, used by other benchmarks (Riochet et al., 2018), is limiting as it requires that anyone wishing to evaluate on INFLEVEL to train a special "surprise" decoder on their model using some external data source. To circumvent this problem, we model surprise as out-of-domain (OOD) detection using the intuition that models with sufficient physical understanding should consider physically implausible events more out-of-domain than physically plausible ones. In Sec. 4, we propose several OOD metrics, each of which takes a representation of a video and returns a scalar quantifying how out-of-domain the video is. While we show, in Sec. 4, that this OOD approach is empirically promising, it is easy to show that there is no surprise metric which can be used to detect physical understanding for all possible models (see App. E.2). Given this, anyone evaluating on INFLEVEL is free to define their own surprise metrics so long as: (1) the same metric is used across all subsets of INFLEVEL (*i.e.* there should not be one metric for Continuity and another for Gravity) and (2) these surprise metrics are neither trained on INFLEVEL nor designed explicitly to exploit regularities in INFLEVEL data (which would be an implicit form of training). Going forward, we hope that researchers will continue to improve and refine the model-agnostic OOD surprise measures we propose.

Now, without loss of generalization, assume that a model has produced a surprise scalar $s_i$ for every video $v_i$ in INFLEVEL-LAB. We would like to aggregate these scores into a single number representing the model's

performance. Surprise scores are often incomparable: if $v_i$ is a video with a sponge and $v_j$ is a video with an apple, $s_i$ may be larger than $s_j$ simply because the model has never seen a sponge during training but has seen apples. To account for this our evaluation is conditional. Recall (Sec. 3.1) that in our Continuity trials, for each primary/secondary object pair $p$ we have four associated trials VV, II, IV, and VI. Let $s_{\text{VV}}(p), s_{\text{II}}(p), s_{\text{IV}}(p)$, and $s_{\text{VI}}(p)$ be the four surprise scores associated with the $p$ pair. Then we compute the aggregate score as

$$\frac{1}{4 \cdot n} \sum_p \sum_{x \in \{\text{VV,II}\}} \sum_{y \in \{\text{IV,VI}\}} \mathbb{1}\{s_x(p) < s_y(p)\} \qquad (1)$$

where the first sum is over the set of all $n$ distinct pairs $p$ in the Continuity set. This quantity can be interpreted as the average accuracy in discriminating between plausible and implausible events. Evaluations for Solidity and Gravity are analogous.

## 4 Experiments

### 4.1 Metrics.

Detecting whether samples are out-of-domain (or, relatedly, outliers) is a challenging problem with a long history (Grubbs, 1969). Recent work (Fort et al., 2021), has shown that very simple OOD metrics, when paired with representations from large-scale models, can result in strong performance in outlier-detection benchmarks. These results suggest that such OOD metrics have the potential to function as a flexible proxy of surprise for deep networks. In INFLEVEL, physically plausible videos are OOD visually (being filmed in a controlled setting from a third perspective) while physically implausible videos are OOD both visually and dynamically (violations of physical principles demonstrate OOD environment dynamics). Thus, for OOD metrics to be useful as a measure of surprise for INFLEVEL

| Metric Aug. | Max-Softmax Sh&PS | NN (L2) Sh&PS | Mahalanobis Sh&PS | von Mises-Fisher Sh&PS |
|---|---|---|---|---|
| None | 0.70 | 0.85 | 0.90 | 0.89 |
| PS | 0.61 | 0.83 | 0.92 | 0.85 |
| Sh | 0.65 | 0.53 | 0.50 | 0.66 |

| Metric Aug. | Max-Softmax Fr&BW | NN (L2) Fr&BW | Mahalanobis Fr&BW | von Mises-Fisher Fr&BW |
|---|---|---|---|---|
| None | 0.69 | 0.85 | 0.57 | 0.84 |
| Fr | 0.69 | 0.56 | 0.49 | 0.78 |
| BW | 0.53 | 0.90 | 0.63 | 0.77 |

Table 1: **Additivity of OOD Metrics.** For four OOD metrics: frequency with which videos with composed column-specified augmentation scored as being *more OOD* than those with the row-specified augmentation. *e.g.* the top left-most entry denotes that 70% of the time the Max-Softmax OOD metric was greater for Sh&PS-augmented videos than for videos with no augmentation. All but the Mahalanobis metric consider the composed augmentations (Sh&PS and Fr&BW) to be more OOD than the others.

where we wish for implausible videos to be *more surprising* than plausible ones, we must have evidence that these metrics are "additive". That is, we wish for these OOD metrics to consider videos with both visual and dynamic abnormalities to be more OOD than those with only visual abnormalities.

To provide evidence for the above, we design the following experiment. Using the popular SlowFast $8 \times 8$ R50 architecture (Feichtenhofer et al., 2019) pretrained on Kinetics400 (Kay et al., 2017), we generate representations on $\approx$16k videos in the Kinetics400 validation set. We then repeat this procedure on six augmented versions of this set: PS, Sh, Sh&PS, BW, Fr, and Fr&BW. PS and BW correspond to visual augmentations (PS swaps patches in individual frames, BW makes the video greyscale), while Sh and Fr correspond to augmentations of dynamics (Sh shuffles video frames, Fr freezes a subset of them). Sh&PS and Fr&BW are then both compositions of visual and dynamic augmentations.

We compute four OOD metrics for each video: Max-Softmax (Hendrycks & Gimpel, 2017), Mahalanobis (Lee et al., 2018), a nearest-neighbor-based metric we call NN (L2), and a metric similar to Mahalanobis but using a von Mises-Fisher distribution in place of the Gaussian. Except for the Max-Softmax metric, these metrics require an additional dataset of videos to be used as in-domain examples. As the above model was trained on Kinetics400, we use the Kinetics400 training set for this purpose. See App. C for more metric details.

Table 1 shows the frequency with which, for each OOD metric, various augmented validation videos are considered more OOD than one another. If an OOD metric is additive, we expect to see that the Sh&PS augmented videos are more OOD than the PS and Sh augmented videos (and similarly for the Fr&BW, BW, and Fr augmentations). This is the trend we observe for all metrics except for the Mahalanobis metric. This provides evidence that OOD metrics can be fruitfully applied in INFLEVEL as a proxy of surprise.

For simplicity, and as the Mahalanobis metric did not consistently show the additive property, we will compute three of the above four metrics across all models in our experiments: the Max-Softmax, NN (L2), and von Mises-Fisher metrics. Moreover, in what follows it will be convenient to be able to aggregate the results from these three metrics in a single number. To this end, we define a "majority vote" (MV) score which, recalling Eq. 1, we compute as

$$\frac{1}{4 \cdot n} \sum_p \sum_{x \in \{VV,II\}} \sum_{y \in \{IV,VI\}} 1\{\text{majority of surprise metrics consider } x \text{ less surprising than } y\} \ . \quad (2)$$

### 4.2 Baselines.

We consider a large collection of video-understanding and embodied AI models. These include: TimeS-Former (TF) (Bertasius et al., 2021) and Motionformer (MF) (Patrick et al., 2021), two recent transformer architectures attaining SoTA performance at action recognition; Channel-Separated Convolutional Networks (CSN) (Tran et al., 2019) and X3D (Feichtenhofer, 2020), two efficient 3D-convolutional video networks achieving high performance across many tasks; Slow (S) / SlowFast (SF) networks (Feichtenhofer et al., 2019), two popular 3D convolution architectures; Two-Stream Inflated 3D ConvNet (I3D) (Carreira & Zisserman, 2017), an early 3D-conv. action recognition model; Contrastive Video Representation Learning (CVRL) network (Qian et al., 2021), a 3D-ResNet-50 backbone trained using a self-supervised contrastive approach achieving impressive performance in the linear-evaluation setting; and finally two embodied AI models trained for the ArmPointNav (Ehsani et al., 2021) and RoboTHOR ObjectNav (Deitke et al., 2020) tasks set in the AI2-THOR simiulator (Kolve et al., 2019), these two models process input images using CNN backbones before passing features to a GRU (Conv2GRU).

### 4.3 Results.

We now describe our results when evaluating the above baseline models on INFLEVEL-LAB and INFLEVEL-SIM. Several of the OOD metrics require reference to a set of videos for, *e.g.*, nearest-neighbor computations. For fair comparison across models, we create a fixed *query set* we call JOINTTRAIN of approximately 25k videos comprised of roughly 5k videos from each of the training splits of the Charades (Ch), HowTo100M (HT100M), Kinetics400 (K400), Something-Something v2 (SSv2), and EpicKitchens100 (EK100) datasets (Sigurdsson et al., 2016; Miech et al., 2019; Kay et al., 2017; Goyal et al., 2017; Damen et al., 2018). In App. E we show that similar results hold when using different query sets.

**INFLEVEL-LAB.** Except for the embodied Conv2GRU models which we train ourselves using (Weihs et al., 2020), see App. F, we use publicly available pretrained models of all baselines. Several of these were available from the PyTorchVideo model zoo (Fan et al., 2021) while others were taken from authors' open-source code. As training dataset may be of importance to success on INFLEVEL, we consider variants of these baselines pretrained on the datasets mentioned above.

Table 2 displays the results for INFLEVEL-LAB using the "majority vote" (MV) score described in Sec. 4.1. See App. Table E.1 for results using individual surprise metrics. For the Solidity and Gravity trials, we see no evidence that these models detect these principles are being violated. While one model (SF-100 pretrained on K400) attains a performance of 0.55 (random performance being 0.50), on the Solidity trials this value is not statistically significant at a 0.01 level. In the Continuity trials, there is a stronger signal: many models score above chance albeit their performance is still quite poor (no model scores above 0.6). This suggests that select models may have begun to form a weak understanding of Continuity. Surprisingly there seems to be little relationship between networks' performance on their training dataset and their performance on INFLEVEL: the Slow-50 models are perhaps the most successful architecture despite consistently under-performing other models. This may suggest that better-performing models overfit to their target domain and learn to rely on domain-specific signals that do not generalize to INFLEVEL. Interestingly, several models have scores that are well below chance implying that they find physically implausible videos *less surprising* than plausible ones. It is tempting to simply negate these model's surprise scores (thus giving above-random performance), we discuss these results further in App. D. In Appendix E.6 we present additional results showing that several models are consistently able to detect differences between trials in INFLEVEL-LAB but

that these models' notion of surprise seems to focus on simple high-level visual attributes (*e.g.* the number of objects visible at the start and end of a video) rather than on the more subtle notion of physical plausibility.

**InfLevel-Sim.** Results on InfLevel-Sim using the MV metric can be found in Table 2 under the InfLevel-Sim heading. See App. Table E.3 for per-surprise-metric results. In-depth discussion is deferred to App. E but we highlight that no models perform at a statistically significant level above chance at Continuity, suggesting that there may be some characteristic difference in how models interpret Continuity in simulation. Several models attain a tiny (maximally 0.51) but statistically significant performance above random chance at Gravity; this is likely an artifact of the number of tests we run simultaneously.

**What violations can these models detect?** The above results suggest that existing models do not have infants' highly general physical-reasoning capabilities. This null result raises two questions: "are our surprise metrics meaningful for the large collection of models under consideration?" and "can we better understand the types of violations that these models can detect?". To better answer these questions, we run an additional collection of experiments similar to those done in Sec. 4.1. In particular, we collect JointEval, a dataset of 5400 videos comprised of approximately 1k videos from evaluation splits of the Ch, HT100M, K400, SSv2, and EK100 datasets. We then create two variants of JointEval by independently applying the Sh (frame shuffling) and PS (patch swapping) augmentations to all videos in JointEval. Finally, we compute surprise scores across all three datasets (original and two variants) and calculate the frequency with which, using the MV score, the various models considered the augmented videos to be more surprising than the originals. The results of these evaluations are given in the last two columns of Table 2.

The results show several surprising trends: (a) Amazingly, the CVRL model is nearly perfect at detecting Sh augmentations suggesting that self-surprised contrastive learning builds a robust understanding of the flow of time; (b) The transformer-based architectures do not consistently consider Sh augmented videos to be surprising, perhaps suggesting that the lack of explicit ordering of inputs in transformer models makes these models less capable in reasoning about time. Interestingly, the TF models generally consider the PS videos to be more surprising than their unaugmented counterparts while the reverse is true for the MF models. (c) Even the untrained SF-50 model performs above chance in detecting the Sh and PS augmented videos demonstrating the power of the inductive biases inherent in 3D convolutional networks. Together, these results provide evidence that our surprise metrics are sufficiently sensitive to give insight into what violations models can detect and allow for meaningful comparisons across models.

## 5 Discussion and Limitations

Our results show that modern AI models lag well behind human infants when reasoning about three core physical concepts. Despite extensive visual experience, existing models struggle to identify violations of these principles despite their ability to accomplish other complex tasks (*e.g.* action recog.). This failure does not imply that the enterprise of building intelligent machines is fruitless. Instead, it implies that new research will be required to build systems that can adequately interpret novel scenarios. Pragmatically, this raises a question, given the growing competencies of artificial agents: **should we care whether our models are capable of physical reasoning?** In the short term, for some "internet AI" settings, the answer is likely no: for a model designed to automate a task that is purely virtual (e.g., recommending videos), a capacity for physical reasoning is of little importance. In the medium to long term, however, we identify two primary reasons why benchmarks like InfLevel are worth taking seriously.

First, while increasingly capable, modern models suffer from a lack of trust due to their reputation as black-box pattern-matchers. InfLevel allows the community to track progress towards physical reasoning and provide evidence that new models form meaningful expectations about the world. If a model cannot reason about basic Continuity violations, we should not trust it to drive a car. Second, physical reasoning appears to be one of the fundamental capabilities (along with *e.g.*, geometry, space, statistical intuition, *etc.*) infants use to bootstrap their growth into general intelligence. Either this reasoning is essential to producing artificial general intelligence (AGI), in which case InfLevel will allow us to track models' progress towards attaining this foundation, or InfLevel will help us illuminate that AGI does not require such reasoning.

We see several promising approaches toward building models that can reason about physical events in novel domains. First, Ha & Schmidhuber (2018) emphasizes the construction of explicit models of the world

| Arch. | Train Set | INFLEVEL-LAB | | | INFLEVEL-SIM | | | JOINTEVAL | |
| | | Continuity | Solidity | Gravity | Continuity | Solidity | Gravity | Sh | PS |
|---|---|---|---|---|---|---|---|---|---|
| CSN | K400 | 0.47 | 0.44* | 0.49 | 0.50 | 0.49 | 0.50 | 0.81* | 0.61* |
| CVRL | K600 | 0.38* | 0.52 | 0.52 | 0.50 | 0.50 | 0.51 | 0.97* | 0.88* |
| Conv2GRU | ArmPN | 0.57* | 0.52 | 0.49 | 0.50 | 0.50 | 0.50 | 0.42* | 0.42* |
| | ObjNav | 0.48 | 0.50 | 0.50 | 0.50 | 0.50 | 0.51 | 0.42* | 0.43* |
| I3D | K400 | 0.50 | 0.50 | 0.51 | 0.50 | 0.50 | 0.51 | 0.69* | 0.56* |
| MF | EK100 | 0.47 | 0.49 | 0.50 | 0.50 | 0.50 | 0.50 | 0.52* | 0.36* |
| | K400 | 0.44* | 0.52 | 0.49 | 0.50 | 0.49 | 0.50 | 0.40* | 0.32* |
| | SSv2 | 0.37* | 0.47 | 0.48 | 0.50 | 0.49 | 0.50 | 0.26* | 0.27* |
| S-50 | Ch | 0.58* | 0.54 | 0.50 | 0.50 | 0.49 | 0.51* | 0.67* | 0.62* |
| | K400 | 0.50 | 0.50 | 0.49 | 0.50 | 0.50 | 0.51* | 0.69* | 0.58* |
| | SSv2 | 0.53 | 0.51 | 0.52 | 0.51 | 0.50 | 0.51* | 0.55* | 0.51 |
| SF-101 | K400 | 0.46* | 0.55 | 0.48 | 0.50 | 0.50 | 0.50 | 0.87* | 0.58* |
| SF-50 | Ch | 0.44* | 0.50 | 0.48 | 0.50 | 0.50 | 0.51 | 0.75* | 0.57* |
| | K400 | 0.49 | 0.51 | 0.51 | 0.50 | 0.50 | 0.51 | 0.87* | 0.57* |
| | None | 0.46* | 0.47 | 0.52 | 0.50 | 0.50 | 0.49 | 0.70* | 0.63* |
| | SSv2 | 0.49 | 0.50 | 0.48 | 0.50 | 0.50 | 0.50 | 0.61* | 0.52 |
| TF | HT100M | 0.45* | 0.48 | 0.49 | 0.50 | 0.50 | 0.50 | 0.51 | 0.78* |
| | K400 | 0.45* | 0.50 | 0.51 | 0.50 | 0.50 | 0.50 | 0.53* | 0.68* |
| | SSv2 | 0.58* | 0.49 | 0.51 | 0.51 | 0.51 | 0.49* | 0.46* | 0.67* |
| X3D | K400 | 0.47 | 0.49 | 0.51 | 0.50 | 0.50 | 0.50 | 0.73* | 0.58* |

Table 2: **Results on INFLEVEL and JOINTEVAL.** Here we display the aggregated "majority vote" score when evaluating various model architectures (Arch.), training datasets (Train Set), and dataset splits (Continuity, Solidity, and Gravity for INFLEVEL and Sh/PS for JOINTEVAL). Several models score above chance ($> 0.5$) on Continuity violations in INFLEVEL-LAB but performance on Solidity and Gravity violations is near-random across the board ($\approx 0.5$). High performance of some models on JOINTEVAL shows that these models can detect less subtle violations which do not require fine-grained object reasoning. $*$ denotes a significant difference from 0.5 at a 0.01 level using a two-sided permutation test.

(*e.g.*, 3D locations of entities and their relationships); reasoning is then performed using these models rather than "model-free" directly from pixels. Evidence suggests that learning generalizable reasoning abilities "model-free" is challenging (Razeghi et al., 2022). A second direction is to work toward incorporating explicit knowledge, such as physical laws, into AI systems (Raissi et al., 2017), possibly using neuro-symbolic techniques (Mao et al., 2019; d'Avila Garcez & Lamb, 2020; Marcus, 2020). A third direction might focus on using richer inductive biases encoding space, time, objects, and causality (Baillargeon & DeJong, 2017; Spelke et al., 1992; Pearl & Mackenzie, 2018)–even in advance of learning.

**Limitations** INFLEVEL is, by design, set in a controlled environment and tests carefully defined components of physical reasoning. Just as infants can become distracted or overwhelmed, even a model that achieves a perfect score on INFLEVEL may struggle to reason in face of diverse, real-world, settings. INFLEVEL provides a valuable diagnostic for physical reasoning and is intended to be used alongside existing benchmarks to provide a holistic perspective on models' fundamental capabilities. As INFLEVEL is evaluation-only, INFLEVEL can only test what a given, frozen, model represents regarding core physical reasoning principles. This means that two models with the same architecture but trained on different data may obtain dramatically different results on INFLEVEL. Thus, when reporting results on INFLEVEL, researchers must take carefully describe their training protocol and avoid attempting to subtly tweak their training dataset to "beat" INFLEVEL; as described in Section 1, as videos in INFLEVEL are tightly controlled, heuristics can be used to "game the system" and obtain high performance but doing so is akin to giving a model the answers to a test: performance is high without understanding. It is also important to emphasize that INFLEVEL is set in the context of an infant-cognition lab and involves household objects. This context may be very different from those used by models during training, *e.g.*, an autonomous driving model will have very little experience with toy ducks but may understand Continuity perfectly when it involves the roadway environment. As above, this suggests a need for a broad spectrum of benchmarks.

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

## Appendices for *Benchmarking Progress to Infant-Level Physical Reasoning in AI*

Our appendices follow below and are organized into the following sections:

A - Additional details regarding our INFLEVEL-LAB dataset.
B - Additional details regarding our INFLEVEL-SIM dataset.
C - Descriptions of the OOD metrics we have used as proxies for surprise during evaluation.
D - A discussion of why negating OOD metric values is not recommended.
E - Additional results and discussion on our evaluation procedure and when varying the query set used during OOD metric computation.
F - Further details on the baseline embodied-AI models we evaluated.
G - Descriptions of assets used in this work (with their associated licenses) and miscellaneous information for INFLEVEL.

## A  INFLEVEL-LAB Details

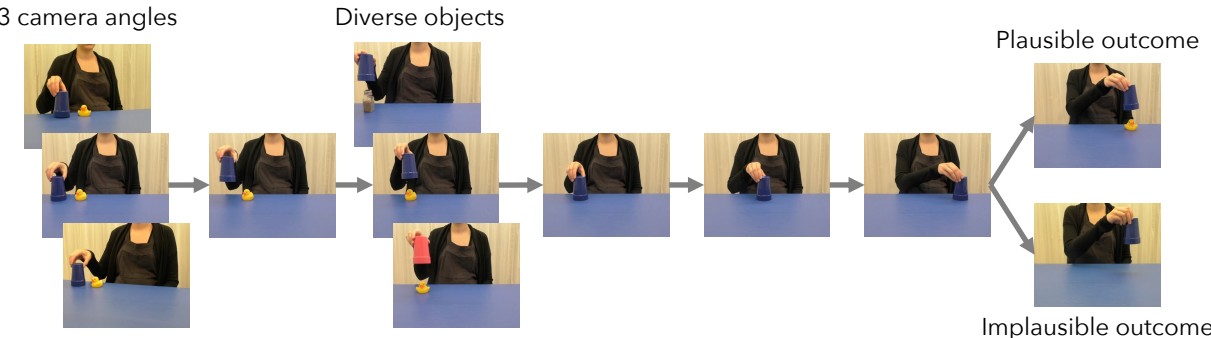

Figure A.1: **Continuity trial**. One Continuity trial from INFLEVEL-LAB (familiarization phases not shown). As illustrated in the first and third columns, we achieve diversity by varying camera angles and using many distinct covers and objects.

Here we provide additional general information about INFLEVEL-LAB videos as well as specific information regarding the Continuity, Solidity, and Gravity trials.

### A.1  General information

As noted in the main paper, all videos were filmed from three camera angles (right, center, and left) at 29.97 FPS in 1280×720 resolution. As is common in infant studies, a metronome was used so that timing across trials was approximately consistent. Primary and secondary objects were sourced from amazon.com, see Fig. 5a for a listing of all secondary object types used during filming. See Fig. A.1 for an example of some of the ways in which we include variety across our videos. Several object types correspond to perishable items (*e.g.* pear, apple, corn, *etc.*), for these items we used non-perishable imitations. We used a single human operator across all videos (an author on this paper). Filming was conducted across multiple days and, to introduce some additional variability into the videos, the operator wore different clothing across days. As trials were filmed over several days and, by their design, have differing image statistics (*e.g.* implausible trials might always show a secondary object while plausible trials only show an secondary object half the time), we apply small video augmentations (jitters of crops, brightness, contrast, saturation, and hue) to all videos.

We film primary and secondary familiarization videos only once and reuse these videos across all relevant trials. For instance, the Continuity videos involving a pepper shaker as the secondary object and a blue cup as the primary object will have the same two familiarization videos as their first two "parts" and will have a

unique final third part (recall Section 3.1 for a description of these parts). As described in the main paper, we concatenate these parts together with short cuts (0.5s). This reuse of familiarization videos has two advantages. First, reusing familiarization videos results in additional data and annotator efficiency. Second, and more importantly, as we use precisely the same habituation videos for the plausible and implausible trials, we remove any potential systematic bias in how the operator handles objects before these trials. We took these shortcuts to correspond similarly to when, in infant trials, a curtain is drawn between the infants and the experimental apparatus.

## A.2 Continuity Data Collection

Recall that we provide two separate physically implausible Continuity trials (see Sec. 3.1). To film these apparently impossible events without advanced visual effects, we employ the following two strategies.

- VI *trial*– Recall that, in this trial, the operator covers a secondary object with a primary object, slides the primary object across the table, and then lifts the primary object to reveal that the secondary object has disappeared.

  To film this trial in practice we discretely attach magnets to the primary and secondary objects. When the primary object is lowered over the secondary object, the magnets engage, and thus, when the primary object is later lifted, the secondary object moves with it and seems to disappear.

- IV *trial*– The operator lowers the primary object onto the table with nothing inside of it, slides the primary object across the table, and then lifts the primary object to reveal that the secondary object has suddenly appeared.

  Filming this trial appears easy initially: we can simply use magnets as above but "in reverse," simply have the object attached by magnets initially and then disengage the magnets when lifting the primary object so that the secondary object seems to magically appear. In practice, we found this approach challenging: disengaging the magnet resulted in using hand motions which made the IV trials systematically different than the other trials.

  Instead, we used another approach: note that a VI trial played in reverse is an IV trial but with the event moving in the opposite direction (*i.e.*, if the operator slid the primary object from left to right, then the reversed video shows an IV trial moving from right to left). So that we did not have a systematic difference between the directions in which the IV and VI trials occur, we filmed two VI trials for each object pair, one in each direction. By reversing these trials, we had a corresponding set of IV trials.

Note that reversing VV and II trials results in new, physically plausible, VV and II trials that, as above, go in the opposite direction (*e.g.* slide from right-to-left if they were originally filmed left-to-right). To match the fact that we include physically implausible videos in both directions, we included these reversed plausible videos in our trials as well.

Now, for every primary/secondary object pair, our videos include 3 camera angles of VV, II, IV, and VI events filmed with the primary object moving left-to-right and with it moving right-to-left. When we compute metrics, we always condition on the object pair as well as camera angle and direction of movement so that surprise score comparisons are fair.

## A.3 Solidity Data Collection

Unlike for the Continuity events above, creating physically implausible videos for our Solidity trials is significantly easier. For every object pair, we film two events, one in which we use the clipped cover (so the secondary object doesn't move with the primary object) and one in which we use the unclipped cover (where the secondary object does move with the primary object). As the unclipped and clipped covers are visually indistinguishable during these trials (the clipped portion, if any, is not visible to the camera) we simply match each of the above two videos with the two possible primary object familiarization videos, *i.e.* one in

which the clipped cover is shown and one in which the unclipped cover is shown. In this way, we generate four videos (2 familiarizations × 2 events) to generate the four (UI, UV, CI, and CV) trials.

For diversity, the direction (left-to-right or right-to-left) that the primary object is moved is varied across videos. Unlike for the Continuity events, however, we do not film left-to-right and right-to-left videos for each object pair.

### A.4   Gravity Data Collection

Our Gravity trials are filmed in a fashion similar to the Solidity trials. As clipped and unclipped containers are, effectively, indistinguishable outside of the familiarization period, we obtain the four trials (UI, UV, CI, and CV) by pairing familiarizations with events (2 familiarizations × 2 events) concatenated with short cuts.

## B   INFLEVEL-SIM Details

Here we provide additional information regarding our simulated INFLEVEL-SIM dataset.

As discussed in the main paper, INFLEVEL-SIM was collected within the AI2-THOR environment (Kolve et al., 2019) using the robotic ManipulaTHOR agent (Ehsani et al., 2021). Just as for INFLEVEL-LAB, we captured videos at 29.97 FPS in 1280×720 resolution from three camera angles. As INFLEVEL-SIM is filmed in simulation we used the fast approximate anti-aliasing algorithm (FXAA) available in the Unity game engine to smooth pixel-based artifacts. Recall Fig. 2 for an example of four frames from a Gravity trial INFLEVEL-SIM. In order to be able to capture our videos of interest, we had to make several changes to AI2-THOR and the ManipulaTHOR agent. In particular, we added actions allowing for additional freedom in arm movement (*e.g.*, rotating the "wrist" and "elbow" of the agent's arm), added a second arm to the agent for use in our Gravity trials (where two objects must be manipulated by the operator simultaneously), and wrote scripts to programmatically "clip" the various AI2-THOR assets (*e.g.* removing the slides or bottoms of cups for the Solidity and Gravity trials). Note that, unlike in App. A, INFLEVEL-SIM being recorded in a simulated environment means that we simply need to change how physics behaves in this environment in order to produce physically implausible videos.

AI2-THOR provides a large collection of object instances from a diverse set of object categories. Figure 5b shows the frequency with which objects of various instances were used across our trials. Moreover, the use of a simulated environment makes it easy to vary the visual aspects of the room in which our videos are set, increasing visual diversity in the dataset. In particular, we randomize (from a curated set of possibilities) the table color/texture (upon which the trial is conducted), background wall color/texture, and the lighting (see Fig. B.2). Note that these augmentations are done in a consistent manner so that all videos using the same primary/secondary object pair will have the same randomizations.

While we have collected INFLEVEL-SIM as a fixed collection of RGB videos, AI2-THOR allows for the easy generation of additional rich data (*e.g.* instance segmentations, depth maps, normal maps, *etc.*). As we will make our data generation scripts publicly available, it will be easy to build upon INFLEVEL-SIM to add any such information as needed. Moreover, we suspect that future models, especially those that are embodied, will perform better when they are allowed to direct their gaze around their environment rather than being forced to view a fixed perspective. This is easy to accomplish in AI2-THOR, and future models that would benefit from such directed gaze will be able to do so in INFLEVEL-SIM.

## C   OOD Metrics

As discussed in the main paper, we consider four metrics for out-of-domain (OOD) detection. We describe these metrics below assuming that we are given a model $f_\theta$ (*e.g.* a video-understanding model) which takes videos $v$ as input. Moreover, we assume that there is some *query* set of videos $Q = \{v_0^q, \ldots, v_N^q\}$ of "in-domain" videos that can be used to compute these metrics (when required).

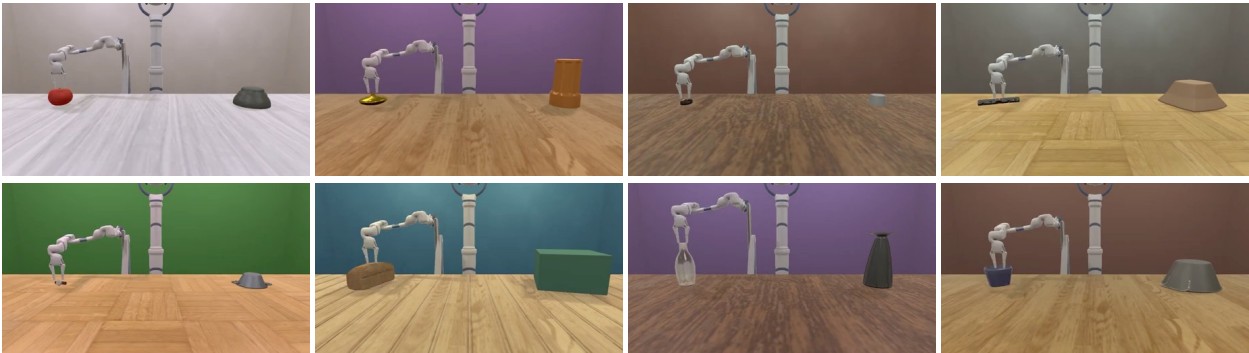

Figure B.2: **Visual diversity in INFLEVEL-SIM** The first frame of 8 randomly selected videos, using the center camera position, in the INFLEVEL-SIM Continuity dataset.

**Max-softmax (Hendrycks & Gimpel, 2017).** The max-softmax (MSM) measure, generally used for classification models, can be applied to any model whose output is a collection of probabilities $f_\theta(v) = (p_0(v), \ldots, p_m(v))$ which sum to one ($\sum_{i=0}^{m} p_i(v) = 1$). The max-softmax measure for such a model and video $v$ then equals

$$\text{Max-Softmax}(f_\theta, v) = \max\{p_0(v), \ldots, p_m(v)\} \quad .$$

The intuition behind this measure is simple: if a video $v$ is in-domain for the model $f_\theta$, then the model should be confident about its predictions and thus the probabilities $p_i(v)$ should be peaked, *i.e.* there should be some $p_i(v)$ near one. Alternatively, if a model is given an OOD video $v$, then it should be less confident about its predictions and the $p_i(v)$ should be near uniform. This simple idea has been shown to be quite effective when paired with deep networks (Fort et al., 2021).

**Mahalanobis (Lee et al., 2018).** Assume now that $f_\theta$ outputs real-valued $d$-dimensional vector representations of input videos $v$. Additionally, assume that we can have some method for assigning every video $v \in Q$ to a single class (or cluster) in $C = \{c_0, ..., c_\ell\}$ so that $\text{cl}(v) \in C$ (*e.g.*, in action recognition, this might be the video's category; if categories don't exist, we can instead use a clustering method such as k-means). The Mahalanobis (MH) OOD measure for a model $f_\theta$ and video $v$ is then computed as

$$\text{MH}(f_\theta, v) = \max_{c \in C} -(f_\theta(v) - \hat{\mu}_c)^T \hat{\Sigma}^{-1} (f_\theta(v) - \hat{\mu}_c)$$

where, for every $c \in C$,

$$\hat{\mu}_c = \text{mean}\{f_\theta(v_i) \mid v_i \in Q \text{ and } \text{cl}(v_i) = c\} \quad \text{and}$$
$$\hat{\Sigma} = \frac{1}{|Q|} \sum_{v_i \in Q} (f_\theta(v_i) - \mu_{\text{cl}(v_i)})(f_\theta(v_i) - \mu_{\text{cl}(v_i)})^T \quad .$$

To understand this intuitively, note that $\hat{\mu}_c$ is a class-conditional mean and $\hat{\Sigma}$ is a $d \times d$ covariance matrix. Thus, up to an additive constant that's fixed across all $c$, the value

$$-(f_\theta(v) - \hat{\mu}_c)^T \hat{\Sigma}^{-1} (f_\theta(v) - \hat{\mu}_c)$$

equals the estimated, $c$-conditional, log-probability of $f_\theta(v)$ when modeling this class-conditional distribution as a multivariate Gaussian with an unknown mean and covariance (this unknown covariance shared across all $c$). Hence $\text{MH}(f_\theta, v)$ is the max of all these log-probabilities and measures the maximum likelihood of $v$ across all classes $c$. With this perspective, the MH metric is quite sensible: videos are considered in-domain when they have a large estimated class-conditional likelihood, and they are considered more out-of-domain when these class-conditional likelihoods are all small.

**von Mises-Fisher**. While the MH measure above models the $c$-conditional distributions as Gaussians, there is no reason, a priori, to assume that the Gaussian distribution is a good fit for this data. We instead propose to model these conditional distributions using the von Mises-Fisher (VMF) distribution, a classical distribution for modeling points on the unit-sphere in high-dimensional spaces (Mardia, 1975). Although we are far from the first to use the VMF distribution in the context of deep-learning (Hasnat et al., 2017), we are, to the best of our knowledge, the first to use the VMF distribution for OOD detection. Our motivation for using the VMF distribution is simple. Many video-understanding models use a single linear layer to map from their representation of a video to their output (*e.g.*, a set of logits used for classification). This linear layer can be interpreted as performing a collection of dot-products between learned vectors and the representation. Up to scaling, this is equivalent to measuring the (cosine of the) angle between the learned vectors and the representation. The VMF distribution uses precisely this type of angle-based reasoning to compute likelihood values, suggesting it may be a more appropriate fit than the standard Gaussian which uses L2 distances.

**Nearest-neighbor-based metric (NN (L2))**. The use of $k$ nearest-neighbors (kNN) for outlier detection is well-studied (Gu et al.). The idea is quite intuitive: a video $v$ is considered in-domain if it is close to (using some measure of distance $D$) other videos in the query set. We compute our NN-based OOD metrics as:

$$\text{NN}_D(f_\theta, v) = -\min_{v^q \in Q} D(v, v^q)$$

where we take $D$ to be the usual L2 distance. Many different candidate metrics can be formed by taking $D$ to be different measures of similarity, e.g. the cosine similarity.

## D   Negating Surprise Scores

As mentioned in the main paper, several models attain performance significantly worse than random upon the Continuity subset of InfLevel-Lab, recall Table 2. For instance, the SSv2-pretrained Motionformer model obtains a score of 0.32 using the NN (CS) measure. It is quite tempting to negate the surprise scores associated with these models, indeed doing so would result in that Motionformer model achieving a score of 0.68, a value 10 percentage points larger than the current best score. We recommend against this approach for two reasons:

1. *Lack of interpretability* - We have used OOD measures as proxies for surprise as they have been well-validated in past work and can be easily interpreted. Indeed, a model that has the capacity for physical reasoning and has trained with physically plausible data should, intuitively, see physically implausible events as out-of-domain. Negated variants of OOD measures have no such interpretation: models should not be rewarded for viewing physically implausible events as more in-domain than physically plausible ones.

2. *Overfitting* - Fair evaluation on InfLevel requires that practitioners *do not* fit their surprise measures to InfLevel. Choosing or modifying surprise measures after observing a model's performance on InfLevel is a form of training on the test set. Suppose, for instance, that we define a surprise measure as $\gamma^T f_\theta(v)$ where $\gamma \in \mathbb{R}^d$ is some vector, $f_\theta$ is a video-understanding model that outputs $d$-dimensional representations, and $v$ is an input video. Then it is easy to see that, by varying $\gamma$ and evaluating the model on InfLevel, we can perform zero-order optimization to find an optimal $\gamma$ maximizing the score. This optimization procedure is, effectively, equivalent to performing logistic regression to discriminate between physically plausible and implausible videos; we describe, in Section 3.4, why this should not be allowed even when using separate training and testing splits.

## E   Additional Results and Discussion

### E.1   Why Shouldn't Linear Decoders Be Used with InfLevel?

Recall the linear-probe evaluation procedure proposed in Section 3.4. Formally, we assume that we are given a trained model $f_\theta$ with parameters $\theta$ that maps an RGB video of dimensions $H{\times}W$ and length $T$ to a

$d$-dimensional vector representation. A common strategy used to measure the quality of representations produced by such a model $f_\theta$ is to freeze the parameters $\theta$ and train a decoder network $g_\gamma$ so that the composed network $g_\gamma \circ f_\theta$ can accomplish some auxiliary task. This evaluation assumes that representations are "good" if they can be used as the foundation for accomplishing many relevant tasks. As noted in the main paper, this evaluation paradigm captures a common use-case for representation learning methods: transfer learning in low-sample regimes. Unfortunately, as the following example shows, using a linear probe to study physical reasoning can result in conflating physical understanding with other related quantities.

**Example** (Conflating features with physical understanding). Consider two functions $S, X : \mathbb{R}^{T \times 3 \times H \times W} \to \mathbb{R}$ where $S$ is a perfect physical-reasoning model (*i.e.* $S(v)$ equals 1 if and only if $v$ is a physically implausible video and $S(v) = 0$ otherwise) and $X$ is a "secondary object tracker", i.e. $X(v) = 1$ if it sees no secondary objects or if it sees the same secondary object at the start and end of $v$ (otherwise equals 0). Now, in our Continuity dataset, training a linear decoder to discriminate plausible and implausible videos will result in high accuracy using either $f_\theta = S$ or $f_\theta = X$ despite the fact that only $S$ has the desired physical understanding. This does not mean that linear evaluation on INFLEVEL is meaningless, the $X$ function described above is a kind of physical-reasoning precursor: identifying and tracking objects is a necessary skill for physical reasoning. Identifying this, however, is not our primary interest.

While this example is somewhat artificial, it demonstrates that allowing training on INFLEVEL can have surprising and unwanted side effects. For this reason, INFLEVEL is an evaluation-only benchmark and we introduce, see Section 3.4, an out-of-domain evaluation procedure that does not require explicit training on INFLEVEL.

### E.2 There is No Universally Applicable Surprise Proxy

We will continue to use the notation introduced in App. E.1. In practice, there is no single surprise proxy which can be used to detect whether or not any given (black box) model contains a well-hidden understanding of core physical reasoning principles. To see this, consider the following pathological example. Suppose that a model that takes a video $v$ and then: (1) computes a hash of the video as $H(v)$, (2) computes the true "plausible or not" boolean $S(v)$, and then (3) finally encrypts the concatenation $[H(v), S(v)]$ using some private key to produce the encrypted representation $E(v)$. Clearly, the model in this pathological example has an understanding of physical plausibility but, unless the surprise proxy is chosen with special knowledge of the private key, this fixed proxy will not be able to detect this from $E(v)$. This suggests that every surprise metric will make some compromises and will not be well suited to all potential models.

### E.3 Per-OOD-Metric Results

Table 2 from the main paper shows results using the "majority vote" score, which aggregates together the three surprise metrics discussed in Section 4.1. See Tables E.1 and E.3 for fine-grained results for each individual surprise metric for INFLEVEL-LAB and INFLEVEL-SIM respectively.

### E.4 Other Query Sets

As described in the main paper, and referenced in App. C, many of our OOD metrics make reference to a query-set $Q$ of in-domain videos. To better compare models, our main paper shows results when using a fixed query set of ≈25k videos from multiple data sources: approximately 5k videos from the training sets from the Charades (Ch), HowTo100M (HT100M), Kinetics400 (K400), Something-Something v2 (SSv2), and EpicKitchens100 (EK100) datasets. We will call this query dataset the *joint query set*.

We also consider pairing each model with a query dataset consisting only of videos from its training dataset, *e.g.*, a model trained on Kinetics400 would have a query set $Q$ consisting only of videos from the Kinetics400 training set. We call these query sets the *train query sets*.[6] The performance of models on INFLEVEL-SIM

---

[6]For our embodied-AI model (trained for the ObjectNav and ArmPointNav tasks in AI2-THOR) there is no static dataset of videos used during training: these models are trained interactively. For this reason, we collect a static dataset of approximately 5k videos in AI2-THOR in which we record the pretrained ArmPointNav model from Ehsani *et al.* (Ehsani et al., 2021) from a third-person perspective as it, in training scenes, completes the ArmPointNav task. We use this collection as our train query

| Arch. | Event Cat. / Metric / Train Set | Continuity | | | Solidity | | | Gravity | | |
|---|---|---|---|---|---|---|---|---|---|---|
| | | MSM | NN (L2) | VMF | MSM | NN (L2) | VMF | MSM | NN (L2) | VMF |
| CSN | K400 | 0.47 | 0.53* | 0.42* | 0.43* | 0.48 | 0.48 | 0.52 | 0.46* | 0.50 |
| CVRL | K600 | - | 0.36* | 0.41* | - | 0.52 | 0.53 | - | 0.51 | 0.53 |
| Conv2GRU | ArmPN | - | 0.58* | 0.56* | - | 0.50 | 0.53 | - | 0.49 | 0.49 |
| | ObjNav | - | 0.49 | 0.47* | - | 0.49 | 0.50 | - | 0.51 | 0.49 |
| I3D | K400 | 0.49 | 0.52 | 0.47* | 0.49 | 0.49 | 0.52 | 0.52 | 0.50 | 0.50 |
| MF | EK100 | 0.53 | 0.48 | 0.46* | 0.52 | 0.51 | 0.48 | 0.53 | 0.50 | 0.49 |
| | K400 | 0.47 | 0.46* | 0.45* | 0.50 | 0.50 | 0.50 | 0.50 | 0.49 | 0.49 |
| | SSv2 | 0.55* | 0.38* | 0.40* | 0.51 | 0.50 | 0.44* | 0.50 | 0.49 | 0.48 |
| S-50 | Ch | 0.49 | 0.56* | 0.60* | 0.52 | 0.52 | 0.52 | 0.50 | 0.50 | 0.49 |
| | K400 | 0.46* | 0.53 | 0.51 | 0.52 | 0.47 | 0.49 | 0.52 | 0.49 | 0.49 |
| | SSv2 | 0.49 | 0.58* | 0.49 | 0.52 | 0.49 | 0.51 | 0.49 | 0.52 | 0.52 |
| SF-101 | K400 | 0.50 | 0.48 | 0.45* | 0.52 | 0.53 | 0.55* | 0.47 | 0.52 | 0.49 |
| SF-50 | Ch | 0.46* | 0.42* | 0.45* | 0.49 | 0.49 | 0.51 | 0.50 | 0.48 | 0.49 |
| | K400 | 0.45* | 0.49 | 0.54* | 0.51 | 0.50 | 0.50 | 0.51 | 0.52 | 0.49 |
| | None | 0.46* | 0.52 | 0.47* | 0.49 | 0.46 | 0.48 | 0.47 | 0.52 | 0.51 |
| | SSv2 | 0.43* | 0.55* | 0.53 | 0.53 | 0.51 | 0.49 | 0.50 | 0.50 | 0.47* |
| TF | HT100M | 0.50 | 0.47 | 0.44* | 0.52 | 0.50 | 0.47 | 0.49 | 0.50 | 0.51 |
| | K400 | 0.51 | 0.45* | 0.44* | 0.52 | 0.49 | 0.51 | 0.50 | 0.51 | 0.51 |
| | SSv2 | 0.55* | 0.52 | 0.59* | 0.50 | 0.48 | 0.51 | 0.51 | 0.50 | 0.50 |
| X3D | K400 | 0.48 | 0.46* | 0.51 | 0.49 | 0.52 | 0.49 | 0.51 | 0.52 | 0.49 |

Table E.1: **Evaluation results on INFLEVEL-LAB using the joint query set.** Compare with E.2.

when using the train query sets is shown in Table E.2. The results are quite similar to those obtained when using the joint query set.

## E.5 Additional Discussion of INFLEVEL-SIM Results

Results of evaluating our various baseline models on INFLEVEL-SIM can be found in Table 2 in the main paper; fine-grained per-surprise-score results are also shown in Tables E.3 (using the joint query set) and Table E.4 (using train query sets). As noted in the main paper, there are two main trends. Firstly, unlike in INFLEVEL-LAB, no models perform significantly above chance at recognizing Continuity violations. This is largely unsurprising for the video-understanding models trained on real videos, as the domain gap between simulated and real videos is quite large. The ObjectNav and ArmPointNav models are, however, trained in simulation and so one might expect these models to do well in this setting. Secondly, several models attain very small (maximally 0.51 with MV and 0.53 with VMF metrics) but statistically significant performances above random chance at detecting Gravity violations. As these values are so small, we hesitate to suggest that this means these models have any understanding of Gravity and are not simply latching onto some statistical correlation present in our data collection methodology.

## E.6 Exploratory Analysis of INFLEVEL-LAB Results

Table E.5 shows an exhaustive breakdown of how frequently each model considers each trial type more surprising than another trial type in INFLEVEL-LAB (using the joint query set and the majority score). This fine-grained breakdown allows us to further analyze the main results presented in Table 2, to better understand the poor performance of models and, in particular, what features of these trials may contribute to model surprise. As these analyses are being done post-hoc, *e.g.* we have already seen model performance from Table E.5 and are now generating intuitively plausible hypotheses by examining the data, this analysis should be seen as exploratory. One such possible hypothesis is that these models simply judge some configurations

set for the embodied models. We will make this dataset open-source and publicly available along with the other components of INFLEVEL.

| Arch. | Event Cat. Metric Train Set | Continuity | | | Solidity | | | Gravity | | |
|---|---|---|---|---|---|---|---|---|---|---|
| | | MSM | NN (L2) | VMF | MSM | NN (L2) | VMF | MSM | NN (L2) | VMF |
| CSN | K400 | 0.47 | 0.52 | 0.42* | 0.43* | 0.47 | 0.49 | 0.52 | 0.49 | 0.51 |
| CVRL | K600 | - | 0.42* | 0.35* | - | 0.53 | 0.52 | - | 0.51 | 0.53 |
| Conv2GRU | ArmPN | - | 0.56* | 0.49 | - | 0.51 | 0.51 | - | 0.49 | 0.49 |
| | ObjNav | - | 0.50 | 0.47* | - | 0.49 | 0.50 | - | 0.52 | 0.49 |
| I3D | K400 | 0.49 | 0.52 | 0.43* | 0.49 | 0.49 | 0.51 | 0.52 | 0.50 | 0.49 |
| MF | EK100 | 0.53 | 0.47 | 0.47 | 0.52 | 0.51 | 0.52 | 0.53 | 0.49 | 0.49 |
| | K400 | 0.47 | 0.45* | 0.41* | 0.50 | 0.49 | 0.49 | 0.50 | 0.49 | 0.50 |
| | SSv2 | 0.55* | 0.40* | 0.35* | 0.51 | 0.52 | 0.49 | 0.50 | 0.48 | 0.50 |
| S-50 | Ch | 0.49 | 0.52 | 0.59* | 0.52 | 0.50 | 0.52 | 0.50 | 0.50 | 0.49 |
| | K400 | 0.46* | 0.54* | 0.47 | 0.52 | 0.48 | 0.49 | 0.52 | 0.50 | 0.50 |
| | SSv2 | 0.49 | 0.57* | 0.54* | 0.52 | 0.50 | 0.53 | 0.49 | 0.52 | 0.53* |
| SF-101 | K400 | 0.50 | 0.49 | 0.48 | 0.52 | 0.53 | 0.53 | 0.47 | 0.52 | 0.50 |
| SF-50 | Ch | 0.46* | 0.47* | 0.43* | 0.49 | 0.51 | 0.50 | 0.50 | 0.48 | 0.48 |
| | K400 | 0.45* | 0.48 | 0.42* | 0.51 | 0.51 | 0.52 | 0.51 | 0.50 | 0.47 |
| | None | 0.46* | 0.51 | 0.45* | 0.49 | 0.52 | 0.50 | 0.47 | 0.54* | 0.49 |
| | SSv2 | 0.43* | 0.52 | 0.46* | 0.53 | 0.48 | 0.50 | 0.50 | 0.50 | 0.50 |
| TF | HT100M | 0.50 | 0.46* | 0.47 | 0.52 | 0.50 | 0.51 | 0.49 | 0.51 | 0.48 |
| | K400 | 0.51 | 0.53 | 0.40* | 0.52 | 0.51 | 0.51 | 0.50 | 0.50 | 0.50 |
| | SSv2 | 0.55* | 0.46* | 0.62* | 0.50 | 0.52 | 0.52 | 0.51 | 0.49 | 0.50 |
| X3D | K400 | 0.48 | 0.40* | 0.49 | 0.49 | 0.52 | 0.49 | 0.51 | 0.50 | 0.50 |

Table E.2: **Evaluation results on INFLEVEL-LAB using train query sets.** Results are quite similar to those using the joint query set from Table E.1.

of objects to be surprising without consideration of the physical plausibility of these configurations. In Table E.6, we show the subset of trial comparisons where we compare only those trials that start and end with both the primary and secondary objects visible (S2E2 trials) against those that start with both objects visible but end with only one visible (S2E1 trials). There are 9 such trial pairs.

Surprisingly, some models are remarkable consistent in finding the S2E2 trials less (or more) surprising than the S2E1 trials on average. For instance, the TF model trained on SSv2 finds the S2E2 trials less surprising than the S2E1 trial across all 9 trial pairs; similar results hold across 8 of 9 pairs for the CVRL and MF+SSv2 models. In contrast, the SF-50 model trained on SSv2 shows opposite results and finds the S2E2 trials more surprising than the S2E1 trials across all 9 pairs. This consistency is remarkable and emphasizes that these models are able to systematically represent differences between INFLEVEL-LAB trials. However, up to our ability to detect, they do not do so in a way that mimics infants' surprise at physical implausibility.

## F    Embodied Models

As noted in the main paper, except for the embodied Conv2GRU models trained for the ObjectNav and ArmPointNav tasks, all baseline models we have evaluated on INFLEVEL come pretrained from external sources. While pretrained models exist for both the ObjectNav[7] and ArmPointNav[8] tasks, we found these baseline models quite challenging to use with INFLEVEL as they were not designed with video-understanding in mind. In particular, the pretrained embodied models have architectures in which, at every timestep, goal information (*e.g.* "you are looking for a cup"), previous action information (*e.g.* "your last action was to move forward"), and egocentric visual inputs (*e.g.* RGB images and depth maps), are each embedded and then passed through an RNN (namely a 1-layer GRU (Chung et al., 2014)) to produce a representation of the environment. Naïvely attempting to use these models in INFLEVEL presents several challenges: (1)

---

[7]https://github.com/allenai/allenact/tree/main/projects/objectnav_baselines
[8]https://github.com/allenai/manipulathor

| Arch. | Event Cat. Metric Train Set | Continuity | | | Solidity | | | Gravity | | |
|---|---|---|---|---|---|---|---|---|---|---|
| | | MSM | NN (L2) | VMF | MSM | NN (L2) | VMF | MSM | NN (L2) | VMF |
| CSN | K400 | 0.50 | 0.50 | 0.50 | 0.50 | 0.50 | 0.50 | 0.50 | 0.49 | 0.50 |
| CVRL | K600 | - | 0.50 | 0.50 | - | 0.50 | 0.50 | - | 0.51 | 0.51 |
| Conv2GRU | ArmPN | - | 0.50 | 0.50 | - | 0.50 | 0.50 | - | 0.50 | 0.51 |
| | ObjNav | - | 0.50 | 0.50 | - | 0.50 | 0.50 | - | 0.51 | 0.51 |
| I3D | K400 | 0.50 | 0.50 | 0.50 | 0.51 | 0.50 | 0.51 | 0.51* | 0.48* | 0.52* |
| MF | EK100 | 0.50 | 0.50 | 0.50 | 0.50 | 0.50 | 0.50 | 0.50 | 0.50 | 0.50 |
| | K400 | 0.50 | 0.50 | 0.50 | 0.50 | 0.50 | 0.50 | 0.50 | 0.50 | 0.50 |
| | SSv2 | 0.50 | 0.50 | 0.49 | 0.50 | 0.49 | 0.50 | 0.50 | 0.50 | 0.50 |
| S-50 | Ch | 0.50 | 0.50 | 0.50 | 0.50 | 0.50 | 0.49 | 0.50 | 0.51 | 0.52* |
| | K400 | 0.50 | 0.50 | 0.50 | 0.50 | 0.50 | 0.50 | 0.50 | 0.49* | 0.52* |
| | SSv2 | 0.51 | 0.50 | 0.50 | 0.49 | 0.51 | 0.50 | 0.50 | 0.49 | 0.53* |
| SF-101 | K400 | 0.50 | 0.50 | 0.50 | 0.51 | 0.50 | 0.50 | 0.50 | 0.49 | 0.50 |
| SF-50 | Ch | 0.50 | 0.50 | 0.50 | 0.50 | 0.50 | 0.50 | 0.50 | 0.51 | 0.51* |
| | K400 | 0.50 | 0.50 | 0.50 | 0.50 | 0.50 | 0.50 | 0.51 | 0.48* | 0.52* |
| | None | 0.50 | 0.50 | 0.50 | 0.50 | 0.50 | 0.50 | 0.50 | 0.50 | 0.49 |
| | SSv2 | 0.50 | 0.50 | 0.50 | 0.50 | 0.50 | 0.50 | 0.50 | 0.47* | 0.53* |
| TF | HT100M | 0.50 | 0.50 | 0.50 | 0.50 | 0.50 | 0.50 | 0.50 | 0.50 | 0.50 |
| | K400 | 0.50 | 0.50 | 0.50 | 0.50 | 0.50 | 0.50 | 0.50 | 0.50 | 0.50 |
| | SSv2 | 0.51 | 0.51 | 0.50 | 0.50 | 0.51* | 0.50 | 0.49* | 0.49* | 0.50 |
| X3D | K400 | 0.50 | 0.50 | 0.50 | 0.50 | 0.50 | 0.50 | 0.50 | 0.49 | 0.50 |

Table E.3: **Evaluation results on INFLEVEL-SIM using the joint query set.** Compare with E.4.

there is no obvious "goal" in INFLEVEL to give to these models, (2) it is the operator who takes actions in INFLEVEL and not the agent hence there is no last action to embed, and (3) while AI2-THOR can provide ground truth depth maps, INFLEVEL is primarily designed to test models' ability to understand RGB videos without ideal geometric information.

For the above reasons, we have modified the baseline architectures provided for the ObjectNav and Arm-PointNav tasks and retrained these models using the AllenAct (Weihs et al., 2020) framework. Our new architectures use only egocentric RGB images and integrate goal / last action information only after the embedding of the RGB image has been passed through the GRU. In this way, the GRU acts as a passive, goal-independent, memory of prior observations, and the representation coming from this GRU can be used for our evaluations. For further details please see our code base, our pretrained models will be made publicly available.

# G  Asset Licenses and Other Details

## G.1  Asset Licenses

We have used multiple assets in this work, we list relevant licenses below.

- Charades dataset[9] - Licensed for non-commercial use.
- HowTo100M dataset[10] - Apache License 2.0.
- Kinetics400 dataset[11] - Creative Commons Attribution 4.0 International License.
- Something-Something v2 dataset (Goyal et al., 2017) - Proprietary dataset of the Twenty Billion Neurons company with non-commercial academic licenses available on request.

---

[9]https://prior.allenai.org/projects/charades
[10]https://www.di.ens.fr/willow/research/howto100m/
[11]https://deepmind.com/research/open-source/kinetics

| Arch. | Event Cat. Metric Train Set | Continuity MSM | NN (L2) | VMF | Solidity MSM | NN (L2) | VMF | Gravity MSM | NN (L2) | VMF |
|---|---|---|---|---|---|---|---|---|---|---|
| CSN | K400 | 0.50 | 0.50 | 0.50 | 0.50 | 0.50 | 0.50 | 0.50 | 0.50 | 0.50 |
| CVRL | K600 | - | 0.50 | 0.50 | - | 0.50 | 0.50 | - | 0.51 | 0.51 |
| Conv2GRU | ArmPN | - | 0.50 | 0.50 | - | 0.50 | 0.49 | - | 0.50 | 0.50 |
| | ObjNav | - | 0.50 | 0.50 | - | 0.50 | 0.50 | - | 0.51* | 0.52* |
| I3D | K400 | 0.50 | 0.50 | 0.50 | 0.51 | 0.50 | 0.50 | 0.51* | 0.48* | 0.53* |
| MF | EK100 | 0.50 | 0.50 | 0.50 | 0.50 | 0.50 | 0.50 | 0.50 | 0.50 | 0.50 |
| | K400 | 0.50 | 0.50 | 0.50 | 0.50 | 0.49 | 0.50 | 0.50 | 0.50 | 0.50 |
| | SSv2 | 0.50 | 0.50 | 0.50 | 0.50 | 0.50 | 0.50 | 0.50 | 0.50 | 0.50 |
| S-50 | Ch | 0.50 | 0.50 | 0.50 | 0.50 | 0.50 | 0.49 | 0.50 | 0.50 | 0.51* |
| | K400 | 0.50 | 0.50 | 0.50 | 0.50 | 0.50 | 0.50 | 0.50 | 0.49* | 0.52* |
| | SSv2 | 0.51 | 0.50 | 0.50 | 0.49 | 0.50 | 0.50 | 0.50 | 0.50 | 0.53* |
| SF-101 | K400 | 0.50 | 0.50 | 0.50 | 0.51 | 0.50 | 0.50 | 0.50 | 0.50 | 0.52* |
| SF-50 | Ch | 0.50 | 0.50 | 0.50 | 0.50 | 0.50 | 0.50 | 0.50 | 0.51 | 0.50 |
| | K400 | 0.50 | 0.50 | 0.50 | 0.50 | 0.50 | 0.49 | 0.51 | 0.48* | 0.51* |
| | None | 0.50 | 0.50 | 0.50 | 0.50 | 0.50 | 0.50 | 0.50 | 0.50 | 0.49 |
| | SSv2 | 0.50 | 0.50 | 0.50 | 0.50 | 0.50 | 0.50 | 0.50 | 0.48* | 0.52* |
| TF | HT100M | 0.50 | 0.50 | 0.50 | 0.50 | 0.50 | 0.50 | 0.50 | 0.50 | 0.51 |
| | K400 | 0.50 | 0.50 | 0.50 | 0.50 | 0.50 | 0.50 | 0.50 | 0.50 | 0.49 |
| | SSv2 | 0.51 | 0.51 | 0.51 | 0.50 | 0.49 | 0.51 | 0.49* | 0.51* | 0.50 |
| X3D | K400 | 0.50 | 0.50 | 0.50 | 0.50 | 0.50 | 0.50 | 0.50 | 0.50 | 0.51 |

Table E.4: **Evaluation results on INFLEVEL-SIM using train query sets.** Compare with E.3.

| Arch. | Trials Real/Magic Train Set | Continuity II-IV R-M | II-VI R-M | II-VV R-R | IV-VI M-M | VV-IV R-M | VV-VI R-M | Solidity CI-UV M-M | CV-CI R-M | CV-UI R-R | CV-UV R-M | UI-CI R-M | UI-UV R-M | Gravity CI-UV M-M | CV-CI R-M | CV-UI R-R | CV-UV R-M | UI-CI R-M | UI-UV R-M |
|---|---|---|---|---|---|---|---|---|---|---|---|---|---|---|---|---|---|---|---|
| CSN | K400 | 0.47 | 0.46 | 0.51 | 0.47 | 0.47 | 0.47 | 0.55 | 0.42 | 0.47 | 0.47 | 0.42 | 0.45 | 0.46 | 0.45 | 0.48 | 0.46 | 0.54 | 0.52 |
| CVRL | K600 | 0.27 | 0.32 | 0.31 | 0.52 | 0.45 | 0.50 | 0.24 | 0.79 | 0.64 | 0.43 | 0.61 | 0.27 | 0.38 | 0.62 | 0.59 | 0.49 | 0.55 | 0.41 |
| I3D | K400 | 0.54 | 0.54 | 0.58 | 0.49 | 0.46 | 0.46 | 0.40 | 0.59 | 0.60 | 0.47 | 0.54 | 0.42 | 0.58 | 0.50 | 0.60 | 0.61 | 0.40 | 0.52 |
| MF | EK100 | 0.44 | 0.44 | 0.43 | 0.50 | 0.48 | 0.52 | 0.53 | 0.52 | 0.50 | 0.46 | 0.48 | 0.50 | 0.43 | 0.57 | 0.52 | 0.48 | 0.51 | 0.43 |
| | K400 | 0.37 | 0.39 | 0.38 | 0.46 | 0.52 | 0.48 | 0.48 | 0.48 | 0.46 | 0.50 | 0.53 | 0.56 | 0.49 | 0.53 | 0.58 | 0.51 | 0.48 | 0.45 |
| | SSv2 | 0.31 | 0.29 | 0.34 | 0.50 | 0.44 | 0.44 | 0.42 | 0.50 | 0.55 | 0.44 | 0.50 | 0.42 | 0.41 | 0.55 | 0.54 | 0.47 | 0.50 | 0.40 |
| Conv2GRU | ObjNav | 0.49 | 0.45 | 0.50 | 0.45 | 0.51 | 0.45 | 0.44 | 0.51 | 0.47 | 0.45 | 0.54 | 0.49 | 0.32 | 0.64 | 0.61 | 0.43 | 0.54 | 0.37 |
| | ArmPN | 0.60 | 0.58 | 0.54 | 0.49 | 0.57 | 0.55 | 0.55 | 0.46 | 0.38 | 0.51 | 0.54 | 0.56 | 0.52 | 0.43 | 0.42 | 0.46 | 0.52 | 0.54 |
| S-50 | Ch | 0.49 | 0.57 | 0.39 | 0.57 | 0.59 | 0.66 | 0.40 | 0.65 | 0.64 | 0.55 | 0.52 | 0.44 | 0.50 | 0.51 | 0.53 | 0.49 | 0.49 | 0.49 |
| | K400 | 0.46 | 0.48 | 0.39 | 0.51 | 0.53 | 0.52 | 0.36 | 0.61 | 0.54 | 0.45 | 0.53 | 0.39 | 0.59 | 0.45 | 0.58 | 0.59 | 0.39 | 0.54 |
| | SSv2 | 0.48 | 0.61 | 0.53 | 0.57 | 0.47 | 0.54 | 0.55 | 0.55 | 0.55 | 0.52 | 0.47 | 0.50 | 0.61 | 0.37 | 0.34 | 0.50 | 0.57 | 0.66 |
| SF-101 | K400 | 0.43 | 0.46 | 0.43 | 0.52 | 0.48 | 0.49 | 0.56 | 0.57 | 0.60 | 0.64 | 0.46 | 0.53 | 0.64 | 0.46 | 0.64 | 0.67 | 0.30 | 0.50 |
| SF-50 | Ch | 0.44 | 0.44 | 0.50 | 0.54 | 0.42 | 0.46 | 0.32 | 0.63 | 0.46 | 0.36 | 0.64 | 0.38 | 0.55 | 0.52 | 0.59 | 0.58 | 0.37 | 0.45 |
| | K400 | 0.49 | 0.47 | 0.50 | 0.46 | 0.54 | 0.47 | 0.48 | 0.58 | 0.59 | 0.56 | 0.45 | 0.46 | 0.57 | 0.51 | 0.58 | 0.61 | 0.40 | 0.52 |
| | SSv2 | 0.53 | 0.48 | 0.54 | 0.47 | 0.48 | 0.48 | 0.64 | 0.36 | 0.41 | 0.50 | 0.52 | 0.64 | 0.69 | 0.33 | 0.39 | 0.54 | 0.41 | 0.64 |
| | None | 0.43 | 0.43 | 0.46 | 0.49 | 0.49 | 0.51 | 0.46 | 0.44 | 0.48 | 0.52 | 0.44 | 0.47 | 0.46 | 0.52 | 0.47 | 0.51 | 0.55 | 0.50 |
| TF | SSv2 | 0.54 | 0.57 | 0.43 | 0.48 | 0.62 | 0.61 | 0.32 | 0.69 | 0.65 | 0.44 | 0.52 | 0.28 | 0.44 | 0.54 | 0.55 | 0.52 | 0.53 | 0.45 |
| | HT100M | 0.40 | 0.42 | 0.42 | 0.49 | 0.49 | 0.50 | 0.51 | 0.43 | 0.39 | 0.43 | 0.55 | 0.53 | 0.19 | 0.60 | 0.32 | 0.21 | 0.78 | 0.38 |
| | K400 | 0.44 | 0.41 | 0.42 | 0.50 | 0.45 | 0.50 | 0.49 | 0.45 | 0.31 | 0.37 | 0.59 | 0.60 | 0.21 | 0.55 | 0.24 | 0.22 | 0.77 | 0.49 |
| X3D | K400 | 0.46 | 0.48 | 0.49 | 0.52 | 0.47 | 0.47 | 0.55 | 0.44 | 0.41 | 0.50 | 0.51 | 0.53 | 0.45 | 0.57 | 0.51 | 0.48 | 0.52 | 0.46 |

Table E.5: **Evaluation results on INFLEVEL-LAB using the joint query set showing all possible comparisons between trials.** Here we show the frequency with which (using the majority score) each model considers one trial type more surprising than another. *e.g.*, in the first column (Continuity, II-IV, R-M) the first value of 0.47 notes that the CSN model considered the II Continuity trial to be more surprising than the corresponding IV trial in 47% of cases. Here R-M, for ease of reference, is used to denote that the first trial type (II) is physically plausible (*i.e.*, Real) whereas the second trial type (IV) is physically implausible (*i.e.*, Magic).

- EpicKitchens100 dataset[12] - Creative Commons Attribution-NonCommercial 4.0 International License.
- AI2-THOR simulator[13] - Apache License 2.0.

---

[12]https://epic-kitchens.github.io/2021
[13]https://github.com/allenai/ai2thor

| Arch. | Train Set | Continuity VV-VI R-M | Solidity UV-CI M-M | CV-CI R-M | CV-UI R-R | UV-UI M-R | Gravity UV-CI M-M | CV-CI R-M | CV-UI R-R | UV-UI M-R | Total > 0.5 |
|---|---|---|---|---|---|---|---|---|---|---|---|
| CSN | K400 | 0.47 | 0.45 | 0.42 | 0.47 | 0.55 | 0.54 | 0.45 | 0.48 | 0.48 | 2 |
| CVRL | K600 | 0.50 | 0.76 | 0.79 | 0.64 | 0.73 | 0.62 | 0.62 | 0.59 | 0.59 | 8 |
| I3D | K400 | 0.46 | 0.60 | 0.59 | 0.60 | 0.58 | 0.42 | 0.50 | 0.60 | 0.48 | 5 |
| MF | EK100 | 0.52 | 0.47 | 0.52 | 0.50 | 0.50 | 0.57 | 0.57 | 0.52 | 0.57 | 7 |
| MF | K400 | 0.48 | 0.52 | 0.48 | 0.46 | 0.44 | 0.51 | 0.53 | 0.58 | 0.55 | 5 |
| MF | SSv2 | 0.44 | 0.58 | 0.50 | 0.55 | 0.58 | 0.59 | 0.55 | 0.54 | 0.60 | 8 |
| Conv2GRU | ObjNav | 0.45 | 0.56 | 0.51 | 0.47 | 0.51 | 0.68 | 0.64 | 0.61 | 0.63 | 7 |
| Conv2GRU | ArmPN | 0.55 | 0.45 | 0.46 | 0.38 | 0.44 | 0.48 | 0.43 | 0.42 | 0.46 | 1 |
| S-50 | Ch | 0.66 | 0.60 | 0.65 | 0.64 | 0.56 | 0.50 | 0.51 | 0.53 | 0.51 | 8 |
| S-50 | K400 | 0.52 | 0.64 | 0.61 | 0.54 | 0.61 | 0.41 | 0.45 | 0.58 | 0.46 | 6 |
| S-50 | SSv2 | 0.54 | 0.45 | 0.55 | 0.55 | 0.50 | 0.39 | 0.37 | 0.34 | 0.34 | 3 |
| SF-101 | K400 | 0.49 | 0.44 | 0.57 | 0.60 | 0.47 | 0.36 | 0.46 | 0.64 | 0.50 | 3 |
| SF-50 | Ch | 0.46 | 0.68 | 0.63 | 0.46 | 0.62 | 0.45 | 0.52 | 0.59 | 0.55 | 6 |
| SF-50 | K400 | 0.47 | 0.52 | 0.58 | 0.59 | 0.54 | 0.43 | 0.51 | 0.58 | 0.48 | 6 |
| SF-50 | SSv2 | 0.48 | 0.36 | 0.36 | 0.41 | 0.36 | 0.31 | 0.33 | 0.39 | 0.36 | 0 |
| SF-50 | None | 0.51 | 0.54 | 0.44 | 0.48 | 0.53 | 0.54 | 0.52 | 0.47 | 0.50 | 6 |
| TF | SSv2 | 0.61 | 0.68 | 0.69 | 0.65 | 0.72 | 0.56 | 0.54 | 0.55 | 0.55 | 9 |
| TF | HT100M | 0.50 | 0.49 | 0.43 | 0.39 | 0.47 | 0.81 | 0.60 | 0.32 | 0.62 | 4 |
| TF | K400 | 0.50 | 0.51 | 0.45 | 0.31 | 0.40 | 0.79 | 0.55 | 0.24 | 0.51 | 5 |
| X3D | K400 | 0.47 | 0.45 | 0.44 | 0.41 | 0.47 | 0.55 | 0.57 | 0.51 | 0.54 | 4 |

Table E.6: **Subset of Table E.5 showing only those comparisons between trials that start with 2 objects visible and end with 2 *v.s.* 1 object visible.** For example, the Continuity VV trial starts and ends with both the primary and secondary objects visible while the Continuity VI trial starts with both objects visible but ends with the primary no longer present. Here, in comparison to Table E.5, we have inverted some scores (new score = $1-$ the old score) so that we are always asking if a model finds the ends-with-two-objects video less surprising than the ends-with-one-object video. We have also added one additional final column that tallies, for each model, the number of times the model obtained a score of >0.5 across the various comparisons. E.g. the TF model trained on SSv2 has all 9 scores >0.5.

- PyTorchVideo library and pretrained models[14] - Apache License 2.0.
- AllenAct framework[15] - MIT License.
- ManipulaTHOR project[16] - MIT License.
- Jacinle library[17] - MIT License.
- Motionformer library and pretrained models[18] - Apache License 2.0.
- SlowFast library[19] - Apache License 2.0.
- "Spatiotemporal Contrastive Video Representation Learning" component of Tensorflow library with associated pretrained model weights[20] - Apache License 2.0.
- PyTorch library[21] - BSD-style license, see here.
- Faiss library[22] - MIT License.
- Timesformer library and pretrained models[23] - Creative Commons Attribution-NonCommercial 4.0 International Public License.

---

[14] https://github.com/facebookresearch/pytorchvideo
[15] https://github.com/allenai/allenact
[16] https://github.com/allenai/manipulathor
[17] https://github.com/vacancy/Jacinle
[18] https://github.com/facebookresearch/Motionformer
[19] https://github.com/facebookresearch/SlowFast
[20] https://github.com/tensorflow/models/tree/master/official/vision/beta/projects/video_ssl
[21] https://github.com/pytorch/pytorch
[22] https://github.com/facebookresearch/faiss
[23] https://github.com/facebookresearch/TimeSformer

### G.2 Other Details

Our INFLEVEL-SIM video dataset was collected by an author on this paper, no other humans were recorded in the creation of any assets produced in this work.

