# OpenReview forum: "Benchmarking Progress to Infant-Level Physical Reasoning in AI"
_TMLR — Accepted by TMLR_

### Review · Reviewer_S66y · 2022-08-05

**Summary Of Contributions:**

This paper proposes InfLevel, a benchmark for physical reasoning in video. The benchmark specifically focuses on three kinds of core knowledge, continuity, solidity, and gravity, which have been studied extensively in humans. The benchmark consists of a large dataset of videos that have violations of expectations under those three principles. The dataset includes both real videos and simulated videos. The benchmark also stipulates that out-of-distribution metrics should be used to evaluate models. In experiments with off-the-shelf models, it is found that all models fail to distinguish the violations of expectation in nearly all cases. Further experiments are conducted to support the design choices.

**Requested Changes:**

- I see no reason why the benchmark proposed here should be limited to "end-to-end learning with deep-learning architectures" -- a benchmark should be approach-agnostic. I would not say that it's necessary to evaluate other kinds of approaches for the present paper, but I do think the manuscript would be strengthened if the mentions of deep learning were replaced with more general terms.
- Relatedly, the introduction briefly mentions that "there is an active community of researchers pursuing [AI models that build in intuitive physics]" and cites Lake et al (2017) as an example. Given the topic of this paper, a much more thorough discussion of models that build in physical priors and take inspiration from cognitive science would be appropriate.

**Strengths And Weaknesses:**

### Strengths

- The paper is overall very rigorous and thorough in its argumentation. I was particularly impressed by the experiments in Section 4.1 to test the hypothesis that surprise is "additive". Many papers would not even articulate that assumption, let alone design an entire experiment to validate it. I was also impressed by the experiments with JointEval, which were helpful for supporting the argument that the main null results are due to weaknesses in existing methods, rather than a flaw in the benchmark itself.
- I like the commitment to "evaluation only" in the benchmark, and found the justification compelling.
- The "Conflating features with physical understanding" example was also convincing and clear. Using OOD instead of training a linear decoder seems like a generally good idea.
- The writing is overall very clear. I only found a few small typos noted below.
- Overall, the paper was a pleasure to read, and I expect that the benchmark will be a useful contribution to the field.

### Weaknesses

- The need for a training set to use with most of the OOD detection methods seems limiting. My main concern is that it could become difficult to disentangle whether the training dataset or the approach under evaluation should be blamed/credited for bad/good OOD detection.
- Against the backdrop of existing work, the real-world version of the benchmark proposed here seems like a much more valuable contribution than the simulated version. I am not convinced that the simulated version is necessary.

---

> ### Author Response · Authors · 2022-08-31
> **Response to Reviewer S66y**
>
> Thank you for your close review and helpful suggestions, we were very happy to hear that you found that "the paper was a pleasure to read, and I expect that the benchmark will be a useful contribution to the field." We respond to your comments below.
>
> #### Weaknesses
>
> **The need for a training set to use with most of the OOD detection methods seems limiting.**
>
> We agree that it would be ideal to have a surprise metric that does not depend on the existence of a query set. Reviewer QbhC brought up a similar concern, we will be sure to add some additional discussion on this topic that care must be taken when comparing models when using different query sets. The need for a query set is driven by our desire to be as model agnostic as possible; as you note, one of our surprise metrics (MSM) does not require a query set but, instead, requires the assumption that the model returns a vector of probabilities as its output. We should note, the Joint query set includes videos from a wide variety of video datasets and so can be seen as being universally applicable.
>
> **Is the simulated data necessary?**
>
> We agree that the InfLevel-Lab is a primary contribution while the InfLevel-Sim can be seen as a secondary contribution. We do believe, however, that InfLevel-Sim provides some exciting opportunities for rapid exploration of tests of other physical principles and for allowing to allow for explicit agent-based interaction (reviewer EhCD, for instance, proposed the interesting idea of taking the robot's perspective so that a model can experiment with the objects before being shown events regarding them).
>
> #### Requested Changes:
>
> **"no reason why the benchmark proposed here should be limited to "end-to-end learning with deep-learning architectures""**
>
> We agree and will make clear that other, non-deep-learning, methods can be evaluated on InfLevel.
>
> **Add further discussion of models using physical priors**
>
> We will add this additional discussion. As reviewer QbhC has asked us to shorten our draft from its current length we will likely need to place this discussion into the appendix. Please let us know if you would strongly prefer this discussion to be in the main text.

---

### Review · Reviewer_QbhC · 2022-08-14

**Summary Of Contributions:**

The paper introduces a new benchmark for evaluating intuitive physical-reasoning capabilities of machine learning models. To this end, the paper focuses on 3 important factors of human physical understanding of short visual sequences - these factors (Continuity, Solidity, and Gravity) have been extensively studied in human infants, and many developmental psychologists even argue that they arise as part of innate core physical-reasoning principles that humans are born with. The benchmark consists of 7.5k clips of real-world recordings (using the same setup that is used to test human infants) from 3 camera angles, and 75k simulated clips representing the same scenes but with a larger variety of objects. The benchmark is intended for evaluation/diagnosis only (of models trained on arbitrary datasets), and is designed to be architecture agnostic. To achieve this, physical understanding of models is measured by quantifying “surprise” (or novelty) on non-plausible scenes compared to plausible scenes (inspired by how physical reasoning is assessed in human infants). Finally, a number of state-of-the-art neural architectures are evaluated on the benchmark, showing (almost) no implicit understanding of intuitive physics, according to the metric evaluated.

To me personally, the idea behind the benchmark is sound and interesting - staying as close as possible to paradigms used in developmental psychology can allow for fruitful exchange between researchers in quite different fields (which is also discussed towards the end of the paper). To me, the dataset seems well designed, and having both a real-world and a simulated part enables interesting research questions (can the simulated data be an adequate replacement for real-world recordings?) and easy scaling to larger and more diverse datasets. My main worry with the current manuscript is the metric that is proposed to assess physical understanding: “surprise” is measured by various distance measures in activation/output space (a technique borrowed from unsupervised outlier detection). A solid and reliable metric is essential to the benchmark, but currently I have some doubts whether the metric is actually capable of reliably measuring what it’s supposed to measure (see more details/discussion below). I currently think that the metric needs more work to be convincing (and I suggest some ideas below) - otherwise the whole paper is reduced to a novel dataset, rather than a benchmark. I am looking forward to hearing the other reviewers’ opinions and authors’ thoughts on this issue.


**Broader Impact Concerns:**

I have no concerns regarding the broader impact.

**Requested Changes:**

Major/important changes

As stated above, there are (at least) three hypotheses that would explain the results observed in Table 1, but only one of the hypotheses is in favor of the main claims of the benchmark results. The other two hypotheses need to be ruled out as much as possible. Below are some suggestions for control experiments, baselines, and ablations (some more, some less elaborate - perhaps not all suggestions are implementable in the relatively short rebuttal phase). I want to encourage the authors to think about additional / more appropriate versions and propose these as well if relevant.

1. The synthetic changes to produce Fig. 1 (PS, Sh, BW, Fr), are quite crude and should lead to clearly distinguishable activation/output patterns because they lead to significant deviations from natural video statistics. The metrics might simply pick up on these deviations. But for the plausible and implausible videos shown used in the benchmark, the high-level video statistics do not change between the two classes. Detecting subtle differences might need different, more sensitive metrics.

1a (simple). Repeat the experiment in Fig. 1 with an untrained network. If the untrained network leads essentially to the same results, then the metric only picks up on input statistics that “leak” into the network activations; which would make the metric less reliable for detecting “surprise” in a model that “understands” what these activation patterns are.

1b (medium). Use the simulator to create two kinds of videos with changes similar to the ones between physically plausible and implausible (i.e. not interfering with natural video statistics), but designed such that FastSlow should be able to pick up on these changes. Repeat the experiment for Fig. 1 with this dataset and show that the metrics proposed work in this setting.

1c (quite elaborate, but most reliable). Train some auto-encoder or other generative model with an explicit latent representation, or a likelihood that can be evaluated on the benchmark data using only the physically plausible part (generate more synthetic data if needed to train the model). Then, measure the likelihood (or latent-space uncertainty) on the non-plausible videos. There should be an increased latent-space uncertainty / decreased likelihood (of generating such a video), corresponding exactly to the desired “surprise” of implausible videos. If this works, then apply the metrics introduced in 4.1 to this model - the metrics should be able to clearly measure the surprise of this model.


2. (simple) As a baseline, train a (linear) detector using the ground-truth data from the benchmark (as suggested at the beginning of Sec. 3.4). Reproduce (parts of) Table 1 with this detector. This is not intended for actual use in the benchmark, but to get an upper bound on the detectability (with all the caveats mentioned in the paper). If this experiment fails, or only delivers poor results, then there is little hope to have more significant results with the metrics proposed in the paper.

2a. Repeat the experiment but with untrained networks - this helps estimate how much of the detector’s accuracy depends on the network actually having learned meaningful representations vs. simply picking up on information that leaks into the network.

3. (simple) What is the impact of the reference dataset (query set) on the metrics? It would be nice to see one or two experiments where the reference dataset is change, just to see whether that has a significant impact on the results or not. This could also be done roughly (but perhaps more easily) via the experiment that leads to Fig. 1.

4. (simple) Subsection “What violations can these models detect?”: the conclusions drawn must be taken with care. The results could equally be explained by having developed a metric that is sensitive to changes in representations caused by PS, Sn, BW, and Fr. To strengthen the results, a comparison against the same evaluation (last two columns in Table 1) but with untrained networks would be interesting.

5. (simple) Briefly clarify in the main text how the metrics in Sec. 4.1 (Table 1) are roughly computed. Do they depend on network outputs only, or do they involve final-layer activations, etc.?

Minor (not crucial for acceptance)

Why is the focus on a single scalar metric? Besides overall accuracy, it could be interesting to report both false positive rates and false negative rates. It seems that most current models would have high false negative rates (implausible video is characterized as not surprising) but low false positive rates (plausible video is characterized as surprising). This could then still be collapsed into something like an area-under-curve metric to get a single scalar.

Why are there short cuts (0.5s) between the phases of the videos? It seems to me that a continuous video would be preferable to assess surprise or intuitive physical understanding. If this is simply a consequence of more data efficiency for recording the videos, this should be noted as such. It would also be good to have at least some discussion why the short cuts are not problematic (after all, infants in the experiment do not get to see these short cuts). The strongest argument would be studies with essentially the same outcomes but with human infants who watch videos instead of a live experimenter and experience the same cuts.

While I appreciate the focus on evaluation only, please discuss briefly that this means that the benchmark results (in the future) must be interpreted with care. The same model trained on different data might easily obtain different benchmark scores. Since the training dataset and protocol is not fixed, future authors reporting results need to take good care of reporting these details, and should avoid tweaking the training data (and perhaps even the reference dataset / query set) to make their model appear to “beat the benchmark”.


**Strengths And Weaknesses:**

**Pro:**
 * Translation of an established and well-studied paradigm in developmental psychology to a setting for testing artificial systems.
 * Novel dataset, consisting of real-world data and simulated data (which can itself enable interesting research on the reliability of pure-simulation evaluation)
 * Well-written paper

**Con:**
 * The metric to assess whether an artificial system understands simple visual physics is to measure “surprise” in non-plausible visual scenes by estimating some distance in activation space and comparing it to the distance observed in plausible visual scenes. Currently no artificial system performs (much) better than chance, which can be due to at least three hypotheses (two of which need to be ruled out for the benchmark to be meaningful):
   * Current ML systems do not have an intuitive understanding of simple visual physics
   * ML systems have an intuitive understanding but the metric is not sensitive enough to pick up on potentially very small “distances”/differences between plausible and implausible activation/output patterns. There is an attempt to verify this in the paper (Sec. 4.1, Fig. 6), but the synthetic changes introduced to make scenes “physically implausible” correspond to large-scale changes to simple statistics which could lead to much more significant “distances” in activation patterns compared to the changes relevant for the benchmark, which might only lead to (small) differences in higher-order statistics (of a fairly high order). Importantly, this hypothesis is still consistent with the results in Table 1 (where the metric is clearly sensitive to the synthetic changes, see last two columns).
   * ML systems have an intuitive understanding and the metric is fine / sensitive enough, but the reference dataset for computing distances (called query set in the paper, Sec. 4.3) is inappropriate. I personally think this is less likely, but nevertheless cannot be taken for granted and needs careful evaluation.

Overall, the only major concern I currently have is w.r.t. the reliability of the proposed metric to actually measure the desired difference. Since the metric is essential to the benchmark, this needs to be strengthened for acceptance of the paper (unless the paper is rewritten as a dataset-only paper, but I personally would not recommend that). Below are some changes and suggestions for doing this - I want to strongly encourage the authors to try and harden the metric as much as possible, because that will ultimately lead to a very meaningful benchmark that will stand the test of time. If this can be achieved, then I think the paper has good potential to be a very strong piece of work.

---

> ### Author Response · Authors · 2022-08-31
> **Response to Reviewer QbhC (1/3)**
>
> Thank you for your in-depth review and your notes that our work is well written and our proposed benchmark is "sound and interesting." We will now respond to your comments and suggestions.
>
> #### **Main concern - "the reliability of the proposed metric to actually measure the desired difference"**
>
> This is an insightful point, we have spent a great deal of time thinking about how one might define a model-agnostic measure of surprise that is both sensitive and does not require retraining; this is a hard problem. Before moving to the suggested approaches for new experiments, we wish to provide some points of context and discussion.
>
> * Demonstrating that our measure of surprise could detect significant deviations from natural video statistics was an important first step in establishing its usefulness; failure to detect such changes would have undermined confidence in our approach. Still, the reviewer is quite right that these synthetic changes are relatively large and crude, and potentially quite different from the more subtle changes between plausible and implausible videos.
>
>   Hopefully addressing some of these concerns, in Section E.4 we present an exhaustive breakdown of how frequently each model considered various trials more, or less, surprising than other trials in InfLevel-Lab. There we see that existing models did distinguish between some trial types. To begin to shed light on what changes the models might be detecting, in Table E.5 we focused on the models' responses (separately within Continuity, Solidity, and Gravity) to trials in which a primary and a secondary object were both visible to start. We then compared the models' responses to pairs of trials, (a) one in which both objects were still visible at the end of the trial vs. (b) one in which only the primary object remained visible at the end of the trial. Across all nine possible comparisons, five systems very consistently judged the (b) videos to be more surprising than the (a) videos, or the reverse, irrespective of the actual plausibility or implausibility of the videos. In another comparison we could include in the revision, we focused on the Solidity and Gravity trials involving (a) an unclipped vs. (b) a clipped primary object. Across all eight possible comparisons, four systems very consistently judged the (b) videos to be more surprising than the (a) videos, or the reverse, irrespective of the actual plausibility or implausibility of the videos. These more consistent systems appear to cluster with respect to model architecture and training dataset, hinting at systematic features of their processing capabilities.
>
>     Though exploratory, these data provide suggestive evidence that our surprise metrics are sensitive enough to detect systematic patterns in the models' responses, even when there are no differences in the videos' high-level statistics. At the same time, they also suggest that the models do not really consider the physically implausible videos to be more surprising, but instead respond to more superficial differences between the videos.
> * In practice, it is impossible to fully rule out that a (black box) model contains a well-hidden understanding of physical principles. A pathological example would be a model that takes a video $v$ and then: (1) computes a hash of the video as $H(v)$, (2) computes the true "plausible or not" boolean $S(v)$, and then (3) finally encrypts the concatenation $[H(v), S(v)]$ using some private key to produce the encrypted representation $R(v)$. Clearly, the model in this pathological example has an understanding of physical plausibility but no model-agnostic surprise metric would be able to detect this from $R(v)$. This simply means that any surprise metric will make some compromises and will not be well suited to all potential models.
>
> * Given the above, we do allow people to define their own surprise metrics when evaluating their models on our benchmark (so long as they do not train these metrics on our data of course) and hope that others will continue to improve and refine our model-agnostic surprise measures.
>
> * Regarding the query set, we should note that the MSM metric uses no query set and the results using the MSM metric are largely consistent with the other metrics (see Tables E.2 and E.3). We hope this provides some additional evidence regarding the (lack of) importance in the query set.

---

> > ### Author Response · Authors · 2022-08-31
> > **Response to Reviewer QbhC (2/3)**
> >
> > #### **Important suggested changes**
> >
> > **Suggestion 1 - Produce a more subtle evaluation of what events can be detected by existing models.**
> >
> > Hopefully addressing some of your concerns, in Section E.4 (see also Table E.5) we describe an exhaustive breakdown of how frequently each model considers various trials more or less surprising than other trials in InfLevel-Lab. There we see that existing models can often very consistently distinguish between different trial types which suggests that our surprise metrics are sensitive enough to detect systematic differences between trials but that the models simply do not consider the physically implausible videos to be the ones that are more surprising.
> >
> > Regarding suggestion 1a. We do show results when using an untrained SF-50 model with the Sh and PS augmentations in Table 1 (see the row with None as a training set and the JointEval column). Here we do see there is some "leakage" as you've described it (the untrained model does find the augmented videos to be more surprising than the unaugmented videos at a level above change) but there is a dramatic difference between its performance and the best performing model which is near perfect in detecting some augmentations. We should also note: there are substantial implicit inductive biases in the design of video understanding architectures like SF-50 regarding the consistent spatial-temporal structure of data even when these models are randomly initialized. Hence the ability of an untrained model to detect Sh and PS augmentations may simply mean that these models are structurally predisposed to detecting such violations. This intuition is supported by work which has shown that untrained convolutional neural networks are surprisingly good feature extractors (see, e.g., Ulyanov, D., Vedaldi, A., & Lempitsky, V. S. (2020). Deep Image Prior. Int. J. Comput. Vis., 128(7), 1867–1888.). Does the above capture the intent behind suggestion 1a or would it be helpful to have further comparisons using the Fr and BW augmentations? These do take some time to generate but I believe we may be able to produce these within the rebuttal period.
> >
> > Regarding suggestions 1b/1c. These are very interesting suggestions, but beyond what we can accomplish in the relatively short rebuttal period. We have considered experiments similar to 1b previously but ran into a problem: it is unclear what these SlowFast models are able to detect. It seems likely that these models are, in some sense, simply tracking changing object textures over time. Following this intuition, we might design a set of videos where object textures change suddenly (related to the Unchangableness property) but a failure of a SlowFast model here would not tell us whether or not our metrics were failing or if our intuition regarding SlowFast's abilities was incorrect. To 1c, our understanding of latent-space methods is that they are not yet ready for "prime time", they work well in some limited settings or when given ground-truth data but defining the likelihood of an arbitrary video is extremely complex. This is an exciting future direction of research.
> >
> > **Suggestion 2 - Linear detector baseline**
> >
> > The primary issue at hand with training a linear detector is one of sample size. All of the video models presented here produce very high-dimensional representations which frequently have larger numbers of dimensions than the number of videos in our dataset. This means that training a linear detector would necessarily overfit, and indeed perfectly fit, the dataset. We do have some thoughts about how we might avoid this overfitting (namely using object-conditional dataset splitting and cross-validation with heavy regularization) which we will further explore; we will report results on this if we find a meaningful experimental setup. This said, the results from Section E.4 seem to strongly suggest that models are able to distinguish between trial types and so it would be surprising if a linear model was not able to pick up on distinction.
> >
> > **Suggestion 3 - Change the query set**
> >
> > We actually do report results using a different query set, see Section E.2 "Other Query Sets". The results change slightly but are largely consistent.
> >
> > **Suggestion 4 - Take care when drawing conclusions in results subsection**
> >
> > Hopefully our results from Section E.4 provide some confidence that our metrics are sensitive to the distinctions between trials in our benchmark. As noted above, we do provide results with one untrained network but will look into providing more untrained model results.
> >
> > **Suggestion 5 - clarify in the main text how the metrics in Sec. 4.1 are computed**
> >
> > We will add this clarification. Other than the MSM metric (which requires probabilities as input), all metrics are computed using the final, non-output, layer from the network.

---

> > > ### Author Response · Authors · 2022-08-31
> > > **Response to Reviewer QbhC (3/3)**
> > >
> > > #### **Minor concerns**
> > >
> > > **Why is the focus on a single scalar metric?**
> > >
> > > We chose to use a single scalar metric, namely accuracy, as it is generally easy to interpret, provides a top-level summary of performance, and we have a clear approach for computing its statistical significance. Note that none of our models actually predict a "plausible" or "implausible" label, instead (see Eq. (2)) we compute whether or not the surprise score for a implausible video is greater than a surprise score for a corresponding plausible video. This approach makes computing false/true positive rates a bit undefined.
> > >
> > > **Why are there short cuts (0.5s) between the phases of the videos?**
> > >
> > > We will add a discussion of the cuts, you are correct that this is due to both data efficiency and precise control over inputs (as it allows us to use exactly the same habituation videos for the plausible and implausible trials). We took these shortcuts to correspond similarly to when, in infant trials, a curtain is drawn between the infants and the experimental apparatus.
> > >
> > > **please discuss briefly that this means that the benchmark results (in the future) must be interpreted with care.**
> > >
> > > Thank you for this suggestion, we will add a discussion of this. Reviewer QbhC has asked us to substantially shorten our draft so this discussion may be somewhat short.

---

> > ### Comment · Reviewer_QbhC · 2022-09-10
> > **Thank you for the detailed response**
> >
> > I want to thank the authors for their detailed response, clarifications of some misunderstanding on my side, and thoughtful walkthrough of relevant results in the Appendix. I am happy to see that the authors have put a lot of thought into the metric, and am particularly pleased to see that the authors essentially agree with many of the difficulties/shortcomings of the current metric. The results in E.4 are interesting, and at least some weak evidence in favor of the proposed metric. While, ideally, I would still like to see some more results regarding the validity of the metric to measure what it's supposed to measure, I think the work as is does pass the threshold for publication - the concerns surrounding the metric remain the weak spot, and I would like to encourage the authors to really try and clear up as much of these concerns as possible, perhaps even in a follow-up project/publication that focuses on the metric(s) only. Together with the dataset, I think that could easily be a significant contribution to the community.

---

### Review · Reviewer_EhCD · 2022-08-23

**Summary Of Contributions:**

The paper introduces 2 new datasets for the evaluation of infant-level physical reasoning in ML systems. It also proposes to use OOD as method to quantify surprise, and experiments with several models showing that their dataset is currently unsolved, despite being trivial to humans.

**Broader Impact Concerns:**

no concerns

**Requested Changes:**

I'm currently recommending acceptance because I believe that the authors did indeed contribute their contributions and I think they'll be used by some members of the ML community.
The main thing's I'd recommend changing though, to make the paper stronger:
- Add at least a comment on your reasoning (cf. W.1).
- Trim the work to 10 pages or less.


**Strengths And Weaknesses:**

### Strengths

- **S.1** The idea of a task that's trivial to humans but currently unsolvable for machine learning systems is great.
- **S.2** The writing and structure are great. It's very accessible.
- **S.3** The supplemental video is long but a great introduction to the proposed dataset.
- **S.4** The plots and graphics are good and illustrative.

### Weaknesses

- **W.1** Importance. Since this is TMLR, I'm evaluating based on if you correctly specified your contributions (you did) and if this might be useful to _someone_ in the ML community (it probably is). So technically, I have to accept the paper. But what I'm worried about is the complete lack of justification as to why these specific tasks are supposed to be great for assessing physical reasoning. You listed 6 other physical-reasoning benchmarks at the end of your related works section and you didn't really mention why continuity, gravity, and solidity are more important than what the others are doing. There's also the work "A roadmap for cognitive development in humanoid robots" by Vernon et al that details a whole range of other capabilities that are also necessary to develop (or test for) human infant-level core knowledge.
- **W.2** Possible future dataset/code release. I generally dislike the statement that the authors will release the dataset/code **after** the paper has been accepted. If your core contribution is a dataset, then release the dataset anonymously (e.g. as link to google drive folder), so the reviewers can evaluate your work.
- **W.3** InfLevel-Sim non-consequitur. You start section 3.3. by saying that there's ample research showing that embodiment and interaction are needed to learn "good" representations and then you go on to introduce a dataset that has none of that. (Also n my humble opinion you missed out on citing Gibson here). Since it's a simulator, what's preventing you from providing an interface to researchers where the simulated agent can poke the test objects on the table or pick them up themselves or even carry out experiments themselves? Just flip the camera to be in the robot's perspective, start with the objects in hand and give the agent the opportunity to move the arm along the x/y/z axes and release either object. Even if this isn't perfect and wouldn't work in all your settings, this would conform to your point that embodiment and interaction lead to a better understanding of a scene.
- **W.4** Choice of objects. See Fig.5 - I would like to call into question that most video models have seen credit cards and soap bars being dropped through funnels. Why did you choose this object frequency? Is this matched to some other (real-world) dataset or is this just based on what was available in AI2-Thor?
- **W.5** Length of the work. I personally find it taxing to go through 12 pages of work and I think nearly every section (and especially the introduction) could have benefitted from being trimmed by a few sentences to a few paragraphs to make them less verbose. I mentioned that the writing is accessible but I think it could be 10 pages and still very clear and enjoyable to read.

### Nitpicks and Misc

- **C.1** Section 3.2 - You can round 29.97FPS to 30 FPS unless the 0.03FPS are critical to your research.
- **C.2** Section 3.4 - The hypothetical here on "why not use a subset of InfLevel" as training dataset was confusing on first read because it sounded like you were actually proposing that.
- **C.3** Section 4.1 - I think the entirety of section 3.4 can be cut from the paper or at least moved to the appendix and this would make space for a brief explanation of the different methods tested in 4.1.
- **C.4** Section 4.2 - I don't see any dedicated intuitive-physics models in your baselines, only video prediction methods. DensePhysNet would be an example or anything trained on the PHYRE benchmark like "Causal World Models by Unsupervised Deconfounding of Physical Dynamics" (Li et al.).
- **C.5** Section 4.3 - I really, really appreciate that you included the significance of your results.
- **C.6** Table.1 - The table appears before you mention what "JointEval" is - so I'd put the table one page later.
- **C.7** Section 5 - Limitation section is great. Discussion is good and should also be cut down to max half its current size because it's way too verbose.

---

> ### Author Response · Authors · 2022-08-31
> **Response to Reviewer EhCD (1/2)**
>
> Thank you for your helpful comments and suggestions, we appreciate that you found our work's foundational idea, writing, video, and plots to be of high quality. We respond to your highlighted weaknesses below.
>
> #### Weaknesses
>
> **W.1 - "...why (are) continuity, gravity, and solidity are more important than what the others are doing"?**
>
> We would not claim that the study of the Continuity, Solidity, and Gravity principles in video reasoning systems is categorically more important than the work done by others. Instead, these principles represent three of the seven core physical-reasoning principles known to be understood by infants (recall paragraph 2 of Section 1). These principles enable infants to correctly interpret a myriad of simple object displacements and interactions, and, as such, they provide a good objective for our research. Here we chose to focus on Continuity, Solidity, and Gravity because they could all be studied with a consistent experimental setup (a single operator manipulating one primary and one secondary object). In future work, we intend to extend our InfLevel to include other principles.
>
> In relation to prior work we note that our work is unique in that it is a real-video benchmark (when most others are purely simulated) that enables studying reasoning systems that _have already been trained for other tasks_ (when others are concerned with training models to accomplish their tasks). As you note, there is a wide range of interesting capabilities that will need to be evaluated before we can claim with confidence that any AI system is approaching parity with infants.
>
> **W.2 - Why not release code and data before acceptance?**
>
> We understand your concern. This choice is largely practical: when releasing our videos we are considering whether or not to anonymize their plausible/implausible status so that it is more challenging to accidentally use this ground-truth information when others evaluate their models. Our existing codebase however assumes that the videos have not been anonymized and so we will need to rewrite these components to use anonymized videos. The code base also requires additional documentation before it will be useable by those unfamiliar with its operation.
>
> Given your desire to evaluate the code and data directly however we will put effort into making at least a portion of the most critical code and data available within this 2 week reponse period.
>
> **W.3 - Lack of embodiment.**
>
> We had actually not considered providing an interface from the robots' perspective, this is a very interesting idea. This is not quite as simple as just flipping to the agent's perspective in some cases as we have "scripted" some object interactions (e.g. when "clipping" the containers/covers our current solution manually ensures that objects can move through these clipped portions). We will look into how we can make this possible (and do intend to release the code we used to flim the simulated videos and so an interested party would always be able to extend our work to add this functionality). We will also add a citation to iGibson.
>
> **W.4 - Why these simulated objects?**
>
> You are correct in that we have used the object categories currently available in AI2-THOR. The distribution seen in Fig. 5 is a result of (1) the number of unique instances available in the various categories and (2) physical constraints induced by the objects (i.e. small secondary objects can fit into more covers/containers and so will have higher representation). We will look into providing a smaller subset of InfLevel-Sim for which the categories are better balanced (a smaller subset will still be more than enough for statistical significance and require a smaller computational burden for those looking to evaluate their models).
>
> **W.5 - Paper is too long**
>
> We will cut down on the verbosity.

---

> > ### Author Response · Authors · 2022-08-31
> > **Response to Reviewer EhCD (2/2)**
> >
> > #### Nitpicks and Misc
> >
> > **C.1 - Round 29.97 FPS to 30 FPS**
> >
> > Before starting this project we were not aware of the subtleties of FPS in recording videos but there is, unfortunately, a meaningful practical difference between 29.97 FPS and 30 FPS that can cause considerable pain when writing code to preprocess videos. To ensure this doesn't happen to others we have leaned to being overly explicit here.
> >
> > **C.2 - Confusing hypothetical**
> >
> > We will make the phrasing more explicit.
> >
> > **C.3 - Cut or move Section 3.4**
> >
> > We will substantially reduce the size of this section and move what remains to the appendix.
> >
> > **C.4 - Dedicated intuitive-physics models baseline**
> >
> > We looked at a number of these as potential baselines but generally have found that they either require ground-truth or non-RGB sensor data, do not have publicly available (or runnable) code, or are not trained using natural videos. For instance, DensePhysNet takes depth maps as input rather than RGB images while PHYRE is a 2D-only benchmark and we suspect that retraining any model designed for that dataset on realistic images would be its own contribution.
> >
> > **C.5 - Appreciate inclusion of statistical significance**
> >
> > Thank you!
> >
> > **C.6 & C.7 - Move Table 1 and reduce limitations section**
> >
> > We will do this.

---

### Comment · Reviewer_EhCD · 2022-08-31
**Author response or discussion between reviewers?**

Heyhey how does this work from here on: do we wait for the authors to respond to our points individually and then we discuss with each other and with the authors? Or do we already discuss with each other?

---

> ### Author Response · Authors · 2022-08-31
> **Re: Author response or discussion between reviewers?**
>
> Hi Reviewer EhCD,
>
> Sorry for the delay, this is our first submission to TMLR and we were also somewhat unclear regarding the timeline for posting the rebuttal. We are finalizing our detailed responses to all reviewer concerns and will post them shortly.
>
> Thank you,
> Authors

---

> ### Comment · Action_Editors · 2022-08-31
> **Clarification of timeline**
>
> Hello, so according to the TMLR guidelines/procedures, as soon as the third review is submitted, there is a discussion phase that is supposed to last 2-4 weeks at which point reviewers should submit final recommendations.  So reviewers and authors can discuss at any time now and until the decision is made.

---

### Author Response · Authors · 2022-09-06
**Revision #1**

We have just submitted our first revision; as it does not seem to be easy to respond to our comments on this revision we will copy these comments below. Please let us know if there are any questions or concerns regarding this revision.

### Code and data
While not a part of the manuscript itself, reviewer EhCD asked us to provide our code and data for evaluation. We provide this below:
 - [Anonymized code](https://anonymous-tmlr22-inflevel.s3.us-west-2.amazonaws.com/inflevel-doubleblind.zip) - This includes our Python codebase along with results files. Please let us know if you have any questions regarding this codebase.
 - [InfLevel-Lab videos](https://anonymous-tmlr22-inflevel.s3.us-west-2.amazonaws.com/lab_merged_256.zip) - This includes all InfLevel-Lab videos resized to have a height of 256 pixels. As described in our response to EhCD, we have obfuscated the path names of these files (their labels) as we have not yet decided if we wish to release this information (as it makes it much easier to game the benchmark).
 - [InfLevel-Sim videos](https://anonymous-tmlr22-inflevel.s3.us-west-2.amazonaws.com/thor_merged_256.zip) - This includes a 3,600-video subset of InfLevel-Sim videos resized to have a height of 256 pixels. We include only a subset of videos here as the full dataset is >20 GB. Please let us know if having the full dataset would be helpful for review.

### Reducing length

We have made substantial cuts to the discussion section and Section 3.4 as requested by reviewer EhCD (see Appendix E.1 for the portion of Section 3.4 that has been moved). These cuts have been somewhat balanced out by requests from other reviewers for additional discussion in other areas (see "minor requested changes" below). We are somewhat hesitant to move Section 3.4 to the appendix entirely as we have found the "evaluation-only" nature of our benchmark to be somewhat controversial and so wish to address it head-on. If there is general consensus among the reviewers that this (or another section) of the paper should be moved to the appendix we would be happy to do this. Otherwise, we will continue to slim down the paper with more minor cuts.

### Minor requested changes

- We have renamed Figure 6 to Table 1.
- [EhCD] We clarified why we study Continuity, Solidity, and Gravity in the introduction.
- [EhCD] We moved Table 2 (previously Table 1) to later in the paper.
- [EhCD] We clarified our hypothetical on "why not to use a subset of InfLevel" in Section 3.4.
- [QbhC] We added a discussion on our choice to use "short-cuts" to Appendix A.1.
- [QbhC] We now include a warning in our Limitations subsection of Section 5 that researchers must take care when reporting their results on InfLevel and should not tweak their training data to "beat the benchmark".
- [S66y] We have de-emphasized "deep-learning" and now reference "AI systems" instead.
- [S66y] We have added additional discussion regarding models using physical priors to the related work.

---

### Public Comment · ~David_Schneider-Joseph1 · 2022-10-04
**Question about VMF score calculation**

Hi, thank you for this very interesting paper.

I have been looking at the computations of surprise scores as implemented in *evaluation/analysis/ood_utils.py* and *evaluation/analysis/surprise_inflevel.py*. I have been able to understand the Max-Softmax, NN (L2), and Mahalanobis surprise calculations.

I have some questions about the computation of the log probability under the Mises-Fisher distribution as implemented in *VonMisesFisherOODModel.score*:

    scores_per_label.append(
        X.shape[1] * 0.5 * np.log(kappa)
        - 0.5 * kappa * (centered * centered).sum(-1)
    )

Because $x$ is normalized earlier in the function, this is equivalent to,

$\frac p 2 \log(\kappa) + \kappa (\mu \cdot x - 1)$,

with $p$ the dimension of the vector space.

However, my calculation for the log probability (ignoring additive constants not dependent on $\kappa$) is:

$(\frac p 2 - 1) \log(\kappa) + \kappa (\mu \cdot x) - \log[I_{p/2 - 1}(\kappa)]$,

where $I_{p/2 - 1}$ is the modified Bessel function of the first kind at order $\frac p 2 - 1$.

Is there some reason I have not understood for the conversion of the term $-\log(\kappa)$ to $-\kappa$, and also the removal of the log of the modified Bessel function (which seems to depend on $\kappa$)?

---

### Decision · Action_Editors · 2022-09-29

**Recommendation:** Accept with minor revision

**Comment:**

The authors present their new evaluation benchmark dataset of videos that are intended to assess physical reasoning of vision-based AI systems.  The benchmark consists of thousands of real videos as well as tens of thousands of synthetic videos of simulated physical environments.  They also propose a set of metrics (and propose pooling them by a voting rule) for assessing vision-based AI systems based whether it correctly identifies videos with physically implausible activity as out-of-domain (conceptually similar to violation-of-expectation / surprise metrics).  These metrics support evaluation of models that have not been trained to output surprise. The authors show that many baselines fail to detect physical violations and they confirm their metrics are sufficiently sensitive under appropriate conditions.

All reviewers believed the work was well motivated and well written.  Two reviewers expressed some concerns about the metrics, which Reviewer QbhC found to be the overall weakest part of the submission in that this reviewer initially expressed some doubts about whether the metric is capable of reliably measuring what it is supposed to measure.

The authors responded thoughtfully and thoroughly to the various reviewer concerns. Prompted by the reviews, the authors shared anonymous code and part of the dataset for reviewer inspection.

Subsequently, all reviewers felt the core contributions were sufficiently strong to accept the submission.  However, Reviewer QbhC continues to believe that the OOD metrics used, which rely on distance between typical plausible and typical implausible inputs, may be overwhelmed by major image statistics and less able to distinguish higher-order statistics associated with physical plausibility vs. implausibility.  The authors do acknowledge limitations of the current metrics in the discussion with the reviewers, though this comes across a little less in the actual paper.  I'd encourage a small clarification somewhere in the paper along the lines of what was communicated in the discussion with Reviewer QbhC.  For example, in the discussion with the reviewer, the authors state "Given the above, we do allow people to define their own surprise metrics when evaluating their models on our benchmark (so long as they do not train these metrics on our data of course) and hope that others will continue to improve and refine our model-agnostic surprise measures."  This open-mindedness could be emphasized in the main text.

typo in the paper: fig 2 "Physicall"

My assessment aligns with the reviewers that this paper should be accepted.

**Audience:**

Yes.

**Claims And Evidence:**

Yes.